# Genetic regulation of serum IgA levels and susceptibility to common immune, infectious, kidney, and cardio-metabolic traits

Immunoglobulin A (IgA) mediates mucosal responses to food antigens and the intestinal microbiome and is involved in susceptibility to mucosal pathogens, celiac disease, inflammatory bowel disease, and IgA nephropathy. We performed a genome-wide association study of serum IgA levels in 41,263 individuals of diverse ancestries and identified 20 genome-wide significant loci, including 9 known and 11 novel loci. Co-localization analyses with expression QTLs prioritized candidate genes for 14 of 20 significant loci. Most loci encoded genes that produced immune defects and IgA abnormalities when genetically manipulated in mice. We also observed positive genetic correlations of serum IgA levels with IgA nephropathy, type 2 diabetes, and body mass index, and negative correlations with celiac disease, inflammatory bowel disease, and several infections. Mendelian randomization supported elevated serum IgA as a causal factor in IgA nephropathy. African ancestry was consistently associated with higher serum IgA levels and greater frequency of IgA-increasing alleles compared to other ancestries. Our findings provide novel insights into the genetic regulation of IgA levels and its potential role in human disease.

Immunoglobulin A (IgA) provides protection against mucosal infections and contributes to the pathogenesis of autoimmune and inflammatory disorders[1,2]. Most of the IgA production occurs at the mucosal surfaces along the gastrointestinal and respiratory tracts, but a large portion of circulating IgA is contributed by bone-marrow plasma cells[3]. IgA neutralizes mucosal pathogens[4] and enhanced IgA responsiveness has been reported in various respiratory and gastrointestinal infections, including acute SARS-CoV-2 infection[5–7]. Increased serum IgA levels are a common phenomenon in patients with IgA nephropathy[8,9], diabetes[10] and metabolic syndrome[11]. The serum concentration of IgA can be influenced by a combination of inherited factors and environmental exposures, including age, sex, and lifestyle factors[11–14]. The heritability of serum IgA levels has been estimated in the range 20–60%[15–18]. Several GWAS have investigated genetic determinants of serum IgA levels in individuals from either European or East Asian ancestry, and nine significant GWAS loci have been identified to date[16,19,20]. Notably, African and other more diverse populations have not been included in prior studies, and limited data exist on the ancestral differences in IgA levels.

In this study, we conducted a genetic analysis of serum IgA levels in 41,263 individuals, including 22,229 diverse participants across 16 ancestry-defined cohorts with genome-wide imputed data combined with 19,034 individuals with summary statistics data on significant and suggestive association signals ($P < 10^{-6}$) from the previous IgA level GWAS by deCODE Genetics and Lund University (deCODE-Lund)[20]. We identified novel genetic determinants of serum IgA levels and, through comprehensive functional annotations, we prioritized candidate causal genes at each of the IgA-associated loci. We then investigated the shared

✉ e-mail: kk473@columbia.edu

genetic architecture between serum IgA levels and other human traits using several approaches, including genome-wide genetic correlation analysis, co-localization of GWAS loci, Mendelian randomization, and phenome-wide association studies. Our results provide new insights into the genetic regulation of serum IgA levels and its role in genetic susceptibility to several human diseases, including immune, infectious, kidney and cardio-metabolic traits.

## Results

### Ancestral differences in serum levels of IgA

We first tested for differences in the distribution of serum IgA levels among 12,260 multi-ancestry individuals broadly classified into 4 major groups based on genetic ancestry (1751 African American, 6791 European, 1257 East Asian, and 2461 Latinx or admixed ancestry individuals). First, we performed laboratory measurements of serum IgA levels in all 5420 participants of the Multi-Ethnic Study of Atherosclerosis (MESA) with available serum samples, providing us with the largest multi-ancestry cohort with standardized IgA measurements. For group comparisons, we generated standardized residuals of log-transformed serum IgA levels after adjustment for age and sex. Notably, MESA participants of African ancestry had significantly higher mean age and sex-adjusted serum IgA levels compared to all other ancestries (Fig. 1a). We next examined the distribution of adjusted IgA levels in the pooled dataset of 6840 diverse non-MESA participants included in our genetic studies. In this independent dataset, we replicated the strong association of genetic African ancestry with higher IgA levels after age and sex adjustment (Fig. 1b). The admixture analysis across the MESA cohort confirmed weak, but highly statistically significant positive correlation between African ancestry and age- and sex-adjusted serum IgA levels ($r = 0.026$, $P = 4.6 \times 10^{-33}$, Fig. 1c). This correlation remained significant after additional adjustment for body mass index (BMI) and diabetes.

### Multi-ancestry GWAS meta-analysis

Next, we aimed to identify genetic loci controlling serum IgA levels using GWAS. We performed a GWAS meta-analysis of 16 diverse ancestry-defined cohorts comprised of 22,229 individuals genotyped genome-wide, combined with 4699 significant and suggestive association signals from an independent cohort of 19,034 North Europeans previously published by deCODE-Lund[20]. Each of the 16 cohorts was genotyped with high-density SNP arrays and imputed using the latest genome sequence reference panels (see Table 1 and Online Methods for details). These cohorts were not ascertained based upon any specific immune or disease phenotype. Within each of the cohorts, IgA phenotypes were defined as standardized residuals of log-transformed IgA levels regressed against age and sex. The final combined dataset comprised of 41,263 individuals with serum IgA measurements (35,094 European, 1751 African American, 1957 East Asian, and 2461 Latinx or admixed-ancestry individuals).

The results of joint meta-analysis are provided in Fig. 2a and Table 2 with regional association plots shown in Supplementary Fig. 1. We observed minimal genomic inflation of the final meta-analysis summary statistics ($\lambda = 1.016$, Supplementary Fig. 2), confirming negligible effects of population stratification. In total, we identified 20 genome-wide significant independent loci, including nine known and 11 novel loci based on $P < 5 \times 10^{-8}$. We detected no significant heterogeneity in associations across all cohorts, and the meta-analysis results were robust under both fixed effects and trans-ancestry (TransMeta[21]) random effects models. Stepwise conditional analyses of the genome-wide significant loci revealed that the HLA locus harbored at least two independently genome-wide significant variants (Supplementary Fig. 3), but no additional independent signals were detected outside of the HLA region. Forests plots in Supplementary Figs. 4, 5 provide detailed comparisons of effect estimates by ancestry for each of the 20 genome-wide significant loci, while Supplementary Data 1 provides the

breakdown of effect estimates and P-values for each individual cohort. We additionally identified eight suggestive signals with $P < 1 \times 10^{-6}$ (Supplementary Table 1), including TNFSF13 locus previously reported in GWAS of Han Chinese ancestry participants[19]. Using linkage disequilibrium (LD)-score regression method[22], we estimated the genome-wide SNP-based heritability of age- and sex-adjusted IgA levels at approximately 7% (95%CI: 2–11%).

As expected, the effect sizes of independently associated variants were inversely related to their minor allelic frequencies (Fig. 2b). Two relatively rare ancestry-specific genome-wide significant variants exhibited the largest effects, including the previously reported RUNX3 locus supported by the European cohorts (IgA-lowering allele rs188468174-T, Beta = −0.88, $P = 3.42 \times 10^{-92}$, European MAF = 1%, absent in African genomes), and the novel GPATCH2 locus supported predominantly by the African ancestry cohorts (IgA-increasing allele rs73100295-T, Beta = 0.36, $P = 3.91 \times 10^{-8}$, MAF = 9% in African ancestry cohorts, MAF = 2% in admixed cohorts, rare in Europeans). Interestingly, the IgA-increasing alleles at 12 of the 20 genome-wide significant loci (60%) were more frequent in African compared to European ancestry, suggesting an enrichment of IgA-increasing alleles in African genomes. To assess if a similar trend was present for all IgA-increasing alleles, we derived a genome-wide polygenic score (GPS) for serum IgA levels (see Online Methods) and compared its distributions between major ancestral populations of the 1000 Genomes Phase 3 reference dataset. The African (AFR) reference population had the highest average GPS of all other ancestral populations (Fig. 1d), with the mean value over two standard deviations higher compared to the European (EUR) population ($t$-test $P < 2 \times 10^{-16}$). The observed distributional differences in the GPS by ancestry had a strikingly parallel pattern to the differences in serum IgA levels (Fig. 1a, b). Because the GPS weights are constant when scoring populations, these observations must be driven by inter-population differences in allelic frequencies. These findings were also consistent with the results of our admixture analysis demonstrating positive correlation between the fraction of African ancestry and IgA levels, leading us to hypothesize that the IgA-increasing alleles could have provided some degree of fitness advantage in Africa, or decreased fitness in non-African environment. To assess for potential polygenic adaptation, we next tested African-European frequency difference for the IgA-increasing alleles for correlation with their corresponding GPS weights (LD-score-adjusted effect sizes assuming 1% fraction of causal variants). We detected a significant positive rank correlation among the top 1% variants with the largest effect (Spearman's $r = 0.01$, $P = 0.0043$, $N = 67,267$ top variants, Fig. 1e). When extended genome-wide by LD score regression[23], this correlation became non-significant ($r_g = 0.05$, $P = 0.32$, $N = 6,710,977$ variants). Similarly, we detected no significant genome-wide correlation by LD score regression between polarized singleton density scores (tSDS)[24] for IgA increasing alleles and their effect sizes ($r_g = -0.07$, $P = 0.31$, $N = 3,635,846$ variants with tSDS scores available).

### Pathway and tissue enrichments

In a protein–protein interaction (PPI) network analysis of positional candidate genes from the non-HLA loci, we observed greater network connectivity than expected by chance (permutation $P = 0.002$, Supplementary Fig. 6), suggesting that the gene products physically interact and thus may participate in common biological processes. Our pathway enrichment analyses based on genome-wide summary statistics[25] identified five significantly enriched pathways at Bonferroni-adjusted $P < 0.05$ (Fig. 3a and Supplementary Table 2), including cytokine signaling in immune system ($P = 1.2 \times 10^{-6}$), signaling by interleukins ($P = 1.1 \times 10^{-5}$), cytokine-cytokine receptor interactions ($P = 5.2 \times 10^{-6}$), TNFs and their physiological receptors ($P = 1.4 \times 10^{-5}$), and IL-6-type cytokine receptor ligand interactions ($P = 5.0 \times 10^{-5}$). Using data-driven expression-prioritized integration for complex traits (DEPICT) analysis[26], we further prioritized 17 tissues and cell types at

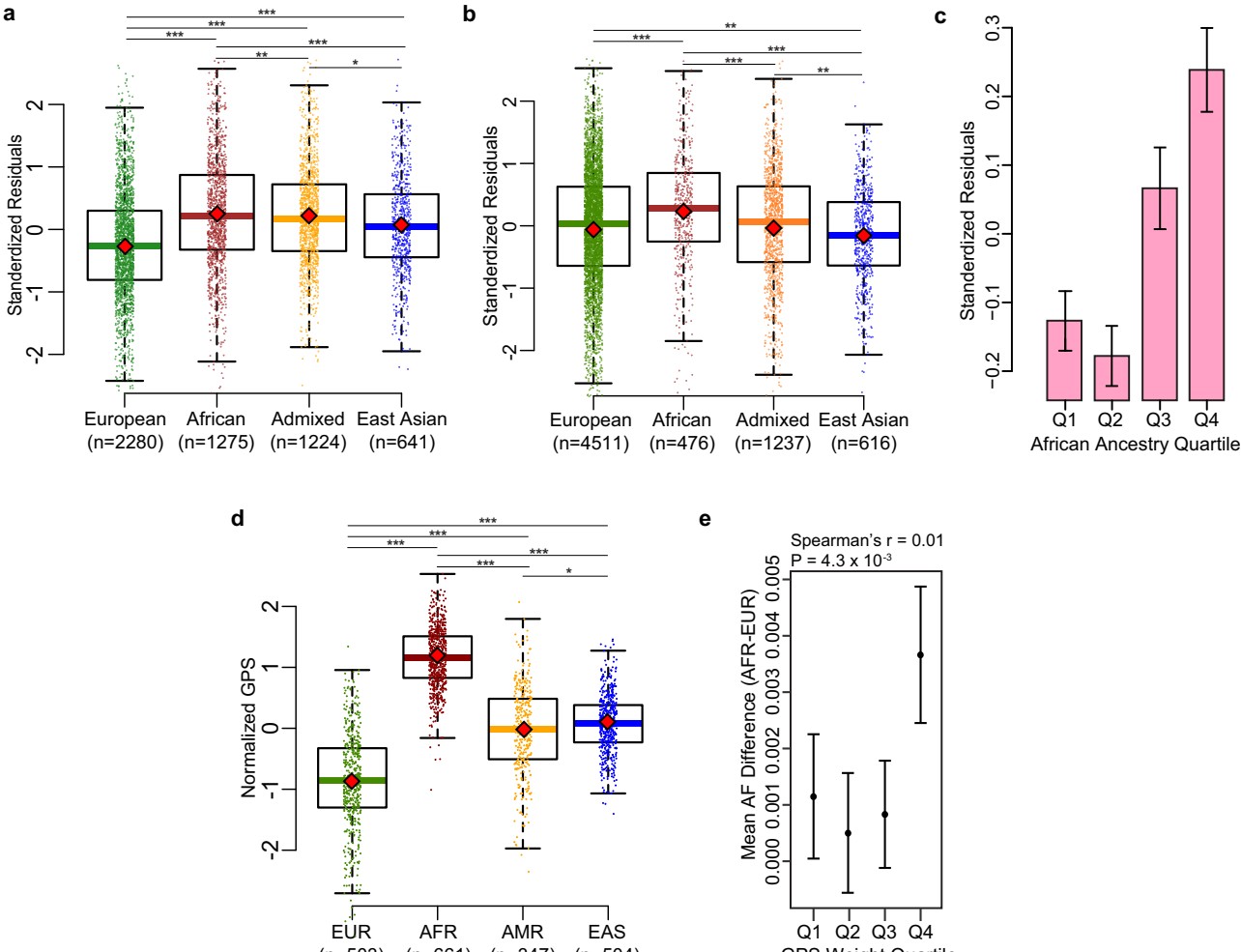

**Fig. 1 | Ancestral differences in serum IgA levels. a** discovery analysis across the four major ancestral groups in MESA demonstrates that African ancestry is associated with higher adjusted IgA levels, **b** replication analysis in all non-MESA study participants confirms higher mean IgA levels in individuals of African ancestry, **c** mean adjusted IgA levels (±95% confidence intervals) as a function of African ancestry fraction, demonstrating that individuals in the upper quartile (>75%) of African ancestry have the highest serum IgA levels ($N$ = 5420 MESA participants); standardized residuals generated by regression of log-transformed serum IgA levels against age and sex were significantly correlated with the African ancestry fraction ($P$ = 4.6 × 10$^{-33}$), and this relationship remained highly significant after additional adjustment for BMI and diabetes ($P$ = 3.7 × 10$^{-23}$), The boxplots in (**a**, **b**, and **d**) depict the median (horizontal line), upper/lower quartiles (boxes), and range (whiskers); the red diamond point denotes the mean value per ancestry group; two-sided unadjusted $t$ test: *$P$ < 0.05; **$P$ < 0.01; ***$P$ < 0.001. **d** distributions of the G$P$S for IgA levels in the 1000 Genomes (Phase 3) populations demonstrating higher GPS in African (AFR) compared to European (EUR), Admixed American (AMR), and East Asian (EAS) populations, **e** mean AFR-EUR difference in IgA-increasing allele frequency for each quartile of GPS weights (LD-score-adjusted effect sizes under the assumption of 0.01 fraction of casual variants, $N$ = 67267 variants included); error bars correspond to 95% confidence intervals around the mean.

FDR < 0.05 based on our GWAS results, with the strongest enrichment in bone marrow cells, hematopoietic system, as well as blood and myeloid cells (Fig. 4b and Supplementary Table 3).

**Pleiotropic associations based on GWAS catalog**

We next interrogated the pleiotropic effect of individual loci by annotating lead SNPs and their proxies against the GWAS catalog database (see Online Methods)[27]. Most of the genome-wide significant loci had previous GWAS associations with immune-mediated disorders, infections, or hematological traits (Fig. 2c and Supplementary Data 2). In particular, eight non-HLA loci, *SH2B3, ANKRD55, HDAC7, RCOR1/TRAF3, TNFSF4, POU2AF1, FADS2/TMEM258,* and *OVOL1/RELA,* displayed either concordant or opposed effects on 18 different autoimmune and inflammatory disorders, suggesting that genetic regulation of IgA levels may play a pervasive role in the control of autoimmunity and inflammation. The *SH2B3* locus displayed the highest degree of pleiotropy, being associated with 79 different GWAS traits. We also found that the alleles associated with higher serum IgA levels at both *SH2B3* and *HORMAD2/LIF* loci were associated with increased risk of tonsillectomy, a procedure frequently performed in the setting of recurrent pharyngitis[28]. Moreover, concordant effects on high blood pressure were found at three loci, including *SH2B3, CTF1* and *HDAC7,* consistent with the epidemiologic association of high blood pressure with increased IgA levels[11]. Interestingly, the *SH2B3* locus showed concordant risk effects on several cardiovascular traits, including coronary artery disease and hypertension, but had an opposed effect on LDL cholesterol. The effect on BMI was concordant with IgA levels for *SH2B3* and *RCOR1/TRAF3* loci, also consistent with the reported correlation between higher serum IgA levels and obesity[11].

**Functional annotations of GWAS loci**

The majority of lead SNPs at genome-wide significant loci map to non-coding (intronic or intergenic) regions (Supplementary Table 4). The lead SNPs at five loci had proxies in the 5′ or 3′UTR regions, in *RUNX2,*

**Table 1 | Baseline characteristics of participants in the GWAS cohorts**

| Cohorts | Ancestry | $N_{Total}$ | $N_{Male}$ | $N_{Female}$ | Mean age |
|---|---|---|---|---|---|
| MESA (European) | European | 2280 | 1132 | 1148 | 64.17 |
| MESA (African) | African | 1275 | 599 | 676 | 64.05 |
| MESA (Admixed 1) | Admixed | 474 | 209 | 265 | 63.38 |
| MESA (Admixed 2) | Admixed | 750 | 369 | 381 | 63.17 |
| MESA (East Asian) | East Asian | 641 | 319 | 322 | 62.48 |
| eMERGE (European) | European | 4261 | 1622 | 2640 | 56.45 |
| eMERGE (African) | African | 476 | 326 | 150 | 50.06 |
| eMERGE (East Asian) | East Asian | 73 | 29 | 44 | 45.38 |
| eMERGE (Admixed 1) | Admixed | 235 | 84 | 151 | 45.63 |
| eMERGE (Admixed 2) | Admixed | 1002 | 427 | 575 | 54.97 |
| German (European) | European | 156 | 104 | 52 | 44.88 |
| French (European) | European | 103 | 30 | 73 | – |
| Chinese (East Asian) | East Asian | 467 | 318 | 149 | 32.39 |
| Japanese (East Asian) | East Asian | 776 | 523 | 252 | 33.79 |
| U.S. (European) | European | 93 | 53 | 40 | 35.66 |
| Swedish (PMID: 24676358) | European | 9167 | 4361 | 4806 | 64.50 |
| deCODE-Lund (PMID: 28628107) | European | 19,034 | – | – | – |
| | All Discovery | 41,263 | 6144 | 6918 | 51.18 |

*ELL2, TNFSF15, TNFSF4, TRAF3*, and *FADS2* genes, and three loci had missense proxies, in *FBXL19, ELL2* and *SH2B3* genes. At the previously reported *ELL2* locus, we identified the Thr298Ala missense variant in *ELL2* exon 7 (rs3815768, $P = 2.9 \times 10^{-27}$) in linkage disequilibrium with the lead SNP (rs3777175, $P = 7.8 \times 10^{-30}$, $r^2 = 0.69$) and located to the end of the ELL2 domain required for transcriptional elongation[29]. At the known *SH2B3* locus, rs3184504 in tight LD with the top SNP ($r^2 = 0.94$) is a missense variant in the canonical transcript of *SH2B3* gene, although this variant is predicted to be benign by PolyPhen2. Given that other top signals map to non-coding regions, we evaluated their potential regulatory function by systematic eQTL co-localization analyses using whole blood[30] and 13 primary immune cell types[31]. The co-localization probability (PP4) exceeded 50% in at least one cell type for 14 of 20 GWAS loci, prioritizing biologically plausible candidate genes at each of these loci (Fig. 4a and Supplementary Table 5). We will next summarize some of our positive functional annotation findings for the top non-HLA GWAS loci.

### RUNX2 and RUNX3 loci
The known *RUNX3* locus on chr.1p36.11 represents one of the strongest signals in our meta-analysis with the largest effect size (rs188468174, Beta = 0.88, $P = 3.42 \times 10^{-92}$). This locus was previously associated with IgG glycosylation[32] and IgA levels[20], and strongly replicated in our study. Interestingly, we have also picked up a novel locus with smaller effect on chr.6p21.1 encoding *RUNX2*, a related transcription factor (rs1200427, Beta = 0.06, $P = 6.85 \times 10^{-14}$). RUNX transcription factors are essential regulators of diverse developmental and signaling pathways[33,34]. Both *RUNX2* and *RUNX3* physically interact[35] and have been linked to retinoic-acid- and TGF-β-induced IgA class switching[36]. The novel *RUNX2* locus co-localized with the eQTL for the *RUNX2* gene in whole blood with PP4 = 0.99; the IgA-decreasing allele at the index SNP (rs1200427-G) was associated with higher mRNA levels of *RUNX2* (Fig. 4a and Supplementary Table 5). Notably, the lead SNP is in LD with rs1200428 ($r^2 = 0.51$), the 3′UTR variant in *RUNX2* that has the strongest eQTL effects in blood. A similar phenomenon was previously reported for the *RUNX3* locus, where the minor allele of the top SNP (rs188468174-T) was associated with lower IgA levels and increased mRNA abundance of the long isoform of *RUNX3*[20]. Therefore, our study

solidifies the evidence for a genetic control model of RUNX transcription factors, in which increased *RUNX* expression suppresses IgA class switching, reducing circulating IgA levels.

### LITAF locus
The new locus on chr.16p13.13 (rs113962704, Beta =0.05, $P = 1.91 \times 10^{-12}$) encodes lipopolysaccharide-induced TNF-alpha factor (LITAF), a transcription factor regulating TNF-alpha expression in intestinal macrophages[37,38]. Notably, macrophage-specific deficiency of LITAF in mice leads to attenuated TNF and IL-6 response upon LPS stimulation[39]. Our analyses revealed that this locus co-localized with eQTL for *LITAF* exclusively in monocytes. Moreover, the top SNP represented a monocyte-specific hQTL (H3K27ac)[40], suggesting that it alters *LITAF* enhancer activity in the monocytic lineage (Fig. 4a, c). The IgA-increasing allele (rs113962704-T) was associated with higher mRNA expression of *LITAF* in monocytes, supporting the hypothesis that this transcription factor provides a stimulus for IgA production by altering monocyte function.

### IL1R1, TRAF3, ANKRD55, and HDAC7 loci
These four loci exhibited T-cell specific eQTL effects. The new locus on chr.2q11.2 (rs13427957, Beta = 0.04, $P = 6.19 \times 10^{-10}$) contains the *IL1R1* gene encoding Interleukin 1 Receptor Type 1, an important T-cell receptor involved in cytokine-induced immune and inflammatory responses[41]. This locus co-localizes with eQTL for *MIR4772* in Th1 cells (PP4 = 0.69) with concordant effect on IgA levels, and *IL1R1* is predicted to be a target of *MIR4772* by miRDB[42] and TargetScan[43]. A mouse knock-out of *IL1R1* in Th2 cells had decreased IgA levels[44].

The new locus on chr.14q32.32 (rs12147883, Beta = 0.05, $P = 5.42 \times 10^{-14}$) contains *TRAF3* encoding TNF Receptor Associated Factor 3, a protein participating in the CD40 signaling, inhibition of non-classical NF-κB signaling, and regulation of class switch recombination in B cells[45–47]. Interestingly, this locus co-localizes with eQTL for *TRAF3* specifically in T cell lineage, where the IgA-increasing variant is associated with higher *TRAF3* expression. In contrast to its inhibitory functions in B-cells, TRAF3 is known to promote many T-cell effector functions through enhancing signaling by the T-cell receptor-CD28 complex[48,49].

Another locus on chr.5q11.2 (rs6859219, Beta = 0.07, $P = 1.41 \times 10^{-20}$) encoding *ANKRD55* (closest gene) and *IL6ST* has previously been associated with IgA levels[20], IgG glycosylation[50] and increased risk of multiple sclerosis, rheumatoid arthritis, and Crohn's disease (Fig. 1c, Supplementary Data 2). Notably, *ANKRD55* gene expression is specific to CD4+ T cells, and mouse studies suggest that ANKRD55 is induced by inflammation and displays T-cell regulatory functions[51]. We co-localized this locus with eQTL for *ANKRD55* in T regulatory cells (PP4 = 0.51) as well as in whole blood (PP4 = 0.98) with concordant effect of *ANKRD55* expression and IgA levels. Moreover, we found that rs6859219 intersects two genetically regulated histone-modification peaks (H3K4me1 and H3K27ac) that are T cell-specific[40] (Supplementary Fig. 7a).

The fourth locus on chr.12q13.11 (rs7487637, Beta = 0.05, $P = 9.97 \times 10^{-15}$) contains *HDAC7* encoding an important histone deacetylase regulating differentiation and function of T-cells[52]. This new locus co-localizes with eQTL for *HDAC7* specifically in T-cells, where the IgA-increasing allele is associated with higher mRNA levels of *HDAC7*.

### OVOL1/RELA locus
The new locus on chr.11q13.1 (rs10896045, Beta = 0.07, $P = 2.57 \times 10^{-22}$) encodes multiple candidate transcripts including *OVOL1*, *RELA*, and several others. The co-localization analysis suggested that this locus was shared with IgA nephropathy, with the IgA-increasing allele associated with increased risk of IgA nephropathy, a kidney disease due to IgA deposition in the glomeruli. Previous GWAS also pointed to the

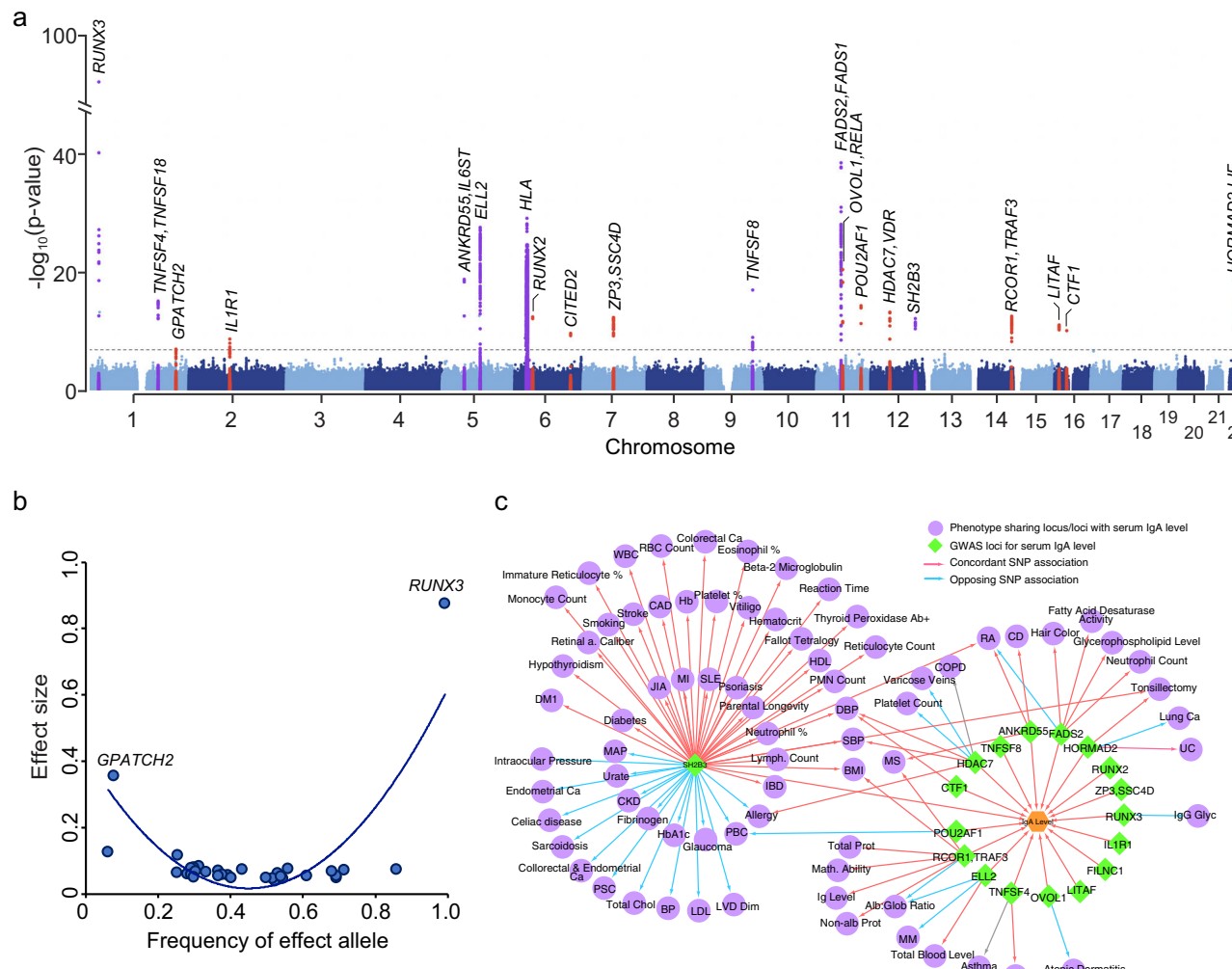

**Fig. 2 | Trans-ethnic meta-analyses across all cohorts identified 20 genome-wide significant loci. a** Manhattan plot depicting a total of 11 novel loci (red) as well as 9 known loci (purple) identified in the meta-analysis; the dotted horizontal line indicates a genome-wide significance threshold ($P = 5 \times 10^{-8}$); the *y*-axis depicts -log10(P-value) for the fixed effects meta-analysis (two-sided, unadjusted), and is truncated to accommodate large peak at *RUNX3* locus. **b** Correlation between average frequency of the independently significant alleles associated with higher IgA levels (x-axis) and their age and sex adjusted effect size (*y*-axis, standardized betas). **c** Pleiotropic effects of the IgA GWAS loci; GWAS loci for IgA levels are in green; other traits are in purple; arrows represent allelic associations that are identical to or in tight LD ($r^2 > 0.5$) with the IgA effect alleles; concordant effects indicated in red; opposed effects in blue.

protective role of this locus from atopic dermatitis (Fig. 2c). While the index SNP localizes to the intronic portion of *OVOL1*, a putative transcription factor of poorly defined function, the nearby *RELA* gene encodes a subunit of NF-κB complex[53]. We found no significant eQTL effects for either *RELA* or *OVOL1* in blood or primary immune cells. However, the lead SNP had significant eQTL effects on *OVOL1* transcript in thyroid ($P = 3.1 \times 10^{-12}$), spleen ($P = 1.5 \times 10^{-6}$) and EBV-transformed lymphocytes ($4.7 \times 10^{-9}$) with the IgA-increasing allele (rs10896045-A) associated with higher expression of *OVOL1* gene in all three GTEx tissues.

### POU2AF1 locus
The new locus on chr.11q23.1 (rs4938518, Beta = 0.056, $P = 7.01 \times 10^{-16}$) encodes *POU2AF1* (POU class 2 homeobox associating factor 1), the gene involved in B-cell antigen responses required for the formation of germinal centers[54]. Mouse knock-out of *POU2AF1* leads to increased B-cell apoptosis and decreased IgA production[55]. The IgA-decreasing allele at this locus has been associated with increased risk of primary biliary cholangitis (Fig. 2c and Supplementary Data 2).

### Additional newly discovered loci
The new locus on 16p11.2 (rs1458201, Beta = 0.052, $P = 2.02 \times 10^{-11}$) contains *CTF1* (encoding cardiotrophin-1) involved in multiple immune-related pathways including cytokine signaling in immune system and interleukins, IL-6-type cytokine receptor ligand interactions and signaling (Fig. 3a). The new locus on chr.6q24.1 (rs17069163, Beta = 0.05, $P = 5.94 \times 10^{-11}$) contains *CITED2*, which encodes a CBP/P300 interacting trans-activator functioning as a molecular switch of TGF-α and TGF-β induced signaling[56], and previously implicated in immune homeostasis and tolerance[57]. The new locus on chr.1q41 (rs73100295, Beta = 0.36, $P = 3.91 \times 10^{-08}$) encodes *GPATCH2*, but its role in the immune system regulation is unknown. Lastly, our co-localization analysis of the new locus on chr.7q11.23 (rs55722505, Beta = 0.048, $P = 8.61 \times 10^{-14}$) suggested several candidate effector genes including *SRCRB4D*, *DTX2* and *YWHAG* in whole blood, *ZP3* and *SSC4D* in monocytes, and *POMZP3* in naïve CD8 and B cells (Fig. 4a).

### Additional known loci validated in this study
We replicated the previously reported *ELL2* locus (rs3777175, Beta = 0.08, $P = 7.81 \times 10^{-30}$), which co-localized with blood eQTL for *ELL2*

**Table 2 | Effect estimates for 20 genome-wide significant loci for IgA levels by trans-ancestry meta-analysis**

| Locus | CHR | BP (hg19) | SNP | Effect Allele | Beta | *P*-value FE | *P*-value RE | I2 | Q | Note |
|---|---|---|---|---|---|---|---|---|---|---|
| *RUNX3* | 1 | 25291697 | rs188468174 | C | 0.876 | 3.42E−92 | 1.37E−91 | 0.0 | 0.55 | Known |
| *TNFSF4, TNFSF18* | 1 | 173163568 | rs7518129 | G | 0.056 | 1.06E−16 | 4.24E−16 | 3.3 | 0.42 | Known |
| *GPATCH2* | 1 | 217563106 | rs73100295 | T | 0.356 | 3.91E−08 | 3.11E−08 | 0.0 | 0.98 | New |
| *IL1R1* | 2 | 102689031 | rs13427957 | C | 0.040 | 6.19E−10 | 3.86E−10 | 33.3 | 0.08 | New |
| *ELL2* | 5 | 95277555 | rs3777175 | G | 0.084 | 7.81E−30 | 3.13E−29 | 0.0 | 0.50 | Known |
| *ANKRD55, IL6ST* | 5 | 55438580 | rs6859219 | C | 0.073 | 1.41E−20 | 4.38E−20 | 12.4 | 0.31 | Known |
| *HLA* | 6 | 31106893 | rs1265094 | A | 0.076 | 1.86E−31 | 7.43E−31 | 37.4 | 0.06 | Known |
| *RUNX2* | 6 | 45526470 | rs1200427 | T | 0.059 | 6.85E−14 | 2.24E−13 | 20.1 | 0.23 | New |
| *CITED2* | 6 | 139975943 | rs17069163 | T | 0.050 | 5.94E−11 | 2.01E−10 | 37.3 | 0.06 | New |
| *ZP3, SSC4D* | 7 | 76034150 | rs55722505 | C | 0.048 | 8.61E−14 | 3.44E−13 | 23.6 | 0.18 | New |
| *TNFSF8, TNFSF15* | 9 | 117692882 | rs3181356 | T | 0.069 | 1.13E−18 | 2.67E−18 | 0.0 | 0.75 | Known |
| *TMEM258, FADS2* | 11 | 61595564 | rs968567 | T | 0.116 | 2.42E−41 | 3.05E−41 | 14.9 | 0.29 | Known |
| *OVOL1, RELA* | 11 | 65555524 | rs10896045 | A | 0.066 | 2.57E−22 | 1.03E−21 | 0.0 | 0.84 | New |
| *POU2AF1* | 11 | 111267394 | rs4938518 | T | 0.056 | 7.01E−16 | 2.80E−15 | 0.0 | 0.74 | New |
| *HDAC7* | 12 | 48214825 | rs7487637 | G | 0.054 | 9.97E−15 | 3.99E−14 | 4.2 | 0.41 | New |
| *SH2B3* | 12 | 111833788 | rs10774624 | G | 0.050 | 1.37E−13 | 3.25E−13 | 0.0 | 0.54 | Known |
| *TRAF3* | 14 | 103239630 | rs12147883 | C | 0.048 | 5.42E−14 | 3.51E−14 | 8.4 | 0.35 | New |
| *LITAF* | 16 | 11717832 | rs113962704 | T | 0.054 | 1.91E−12 | 6.02E−12 | 0.0 | 0.54 | New |
| *CTF1* | 16 | 30916129 | rs1458201 | A | 0.052 | 2.02E−11 | 8.07E−11 | 25.1 | 0.16 | New |
| *HORMAD2, LIF* | 22 | 30448399 | rs193473 | A | 0.067 | 1.58E−20 | 6.30E−20 | 0.0 | 0.51 | Known |

Beta: per allele effect estimate from linear regression expressed in age, sex, and ancestry-adjusted standard deviation units of IgA distribution. The last column indicates known (previously reported) and new (newly discovered) loci. Gene names indicated in italics.

*FE* fixed effects meta-analysis, *RE* random effects (TransMeta) meta-analysis, *I2* Heterogeneity Index, *Q* Cochrane's heterogeneity test *P*-value.

(PP4 = 0.53, Fig. 4a and Supplementary Table 5). Consistent with previous reports[58], the IgA-increasing allele (rs3777175-G) was associated with lower transcript level of *ELL2* ($P = 6.76 \times 10^{-24}$), prioritizing *ELL2* (encoding elongation factor for RNA polymerase II 2) as the most likely causal gene at this locus. The known *TMEM258/FADS2* locus (rs968567, Beta = 0.12, $P = 2.42 \times 10^{-41}$) co-localized with several candidate genes, including *FADS1* and *FADS2* (fatty acid desaturases regulating unsaturation of fatty acids, co-localized across most blood cell types) and the *TMEM258* gene (co-localized specifically in T cells), with the IgA-increasing allele (rs968567-T) associated with higher expression of all three transcripts. Notably, rs968567 falls into a genetically regulated histone-modification peak (H3K4me1) for *FADS2* in T cells, monocytes, and neutrophils[40] (Supplementary Fig. 7b). *TMEM258* represents a particularly attractive candidate gene at this locus, since mice deficient in *TMEM258* exhibit severe intestinal inflammation[59]. We additionally replicated three known loci encoding members of tumor necrosis factor ligand superfamily; *TNFSF4/TNFSF18* locus on chr.1q25.1[20] (rs7518129, Beta = 0.06, $P = 1.06 \times 10^{-16}$, *TNFSF4* is the closest gene), *TNFSF8/TNFSF15* locus on chr.9q33.1[16] (rs3181356, Beta = 0.07, $P = 1.13 \times 10^{-18}$, *TNFSF8* is the closest gene), and a suggestive *TNFSF13* locus on chr.17p13.1[19] (rs3803800, Beta = 0.06, $P = 9.41 \times 10^{-08}$). These loci encode powerful TNFSF cytokines with partially overlapping receptors[60]. *TNFSF8/TNFSF15* and *TNFSF13* loci are also associated with the risk of IgA nephropathy, while *TNFSF4/TNFSF18* locus has previously been associated with the risk of eczema, asthma and narcolepsy, all with concordant effects to IgA levels (Fig. 2c and Supplemental Table 5).

**Convergence with relevant mouse phenotypes**
To further prioritize causal genes at our GWAS loci, we tested for significant gene set overlaps with human orthologs that cause immune-related phenotypes when disrupted in mice. This included gene sets for abnormal IgA levels (120 genes), abnormal immune tolerance (408 genes) and abnormal response to infection (562 genes) from the Mouse Genome Informatics (MGI) database[55]. We identified

significant overlap for 13 genes linked to abnormal IgA levels in mice (enrichment $P = 5.8 \times 10^{-5}$), 29 linked to abnormal immune tolerance (enrichment $P = 6.3 \times 10^{-6}$) and 42 linked to abnormal response to infection (enrichment $P = 1.1 \times 10^{-5}$) (Fig. 3b and Supplementary Tables 6–8). Among the 120 genes with abnormal IgA phenotypes in mice, six additional genes (*TRIM13, NCR1, IRF5, FGR, ARHGAP15* and *REL*) surpassed a Bonferroni-corrected threshold ($P = 0.05/22169 = 2.26 \times 10^{-06}$) when tested against our GWAS results; these represent plausible new candidate genes for regulation of IgA production in humans.

**Relationship to the risk of IgA nephropathy and tonsillectomy**
Given the known role of the IgA system in the pathogenesis of IgA nephropathy, a kidney disease caused by glomerular deposition of IgA in the setting of pharyngitis and other mucosal infections, we specifically tested for shared genetic architecture between IgA levels, IgA nephropathy, and tonsillectomy[61] by systematic lookups and co-localization analyses of genome-wide significant loci. Among all 20 independent loci associated with higher IgA levels, 8 had nominal associations with increased risk of IgA nephropathy ($P < 0.05$), all with concordant effects (Supplementary Table 9). Among these, there were five genome-wide significant loci with high probability of shared causal variants (PP4 > 0.7), *TNFSF4/TNFSF18, ANKRD55/IL6ST, OVOL1/RELA, SH2B3*, and *HORMAD2/LIF* (Supplementary Table 10). There were also four loci with an overlapping genomic position, but high probability of different causal variants between the two traits (PP3 > 0.7), *HLA, CTF1, TRAF3*, and *TNFSF8/TNFSF15*. Between IgA levels and tonsillectomy, we observed two co-localized loci (*SH2B3* and *HORMAD2/LIF*), both with concordant effects, and another two loci (*HLA* and *CTF1*) with high probability of different causal variants (Supplementary Table 11). Remarkably, the *HORMAD2/LIF* locus was genome-wide significant in all three GWAS and co-localized across all three traits, suggesting a common genetic mechanism (Fig. 5a, b).

To further explore the genetic relationships between these traits, we performed two sample Mendelian Randomization (MR) between

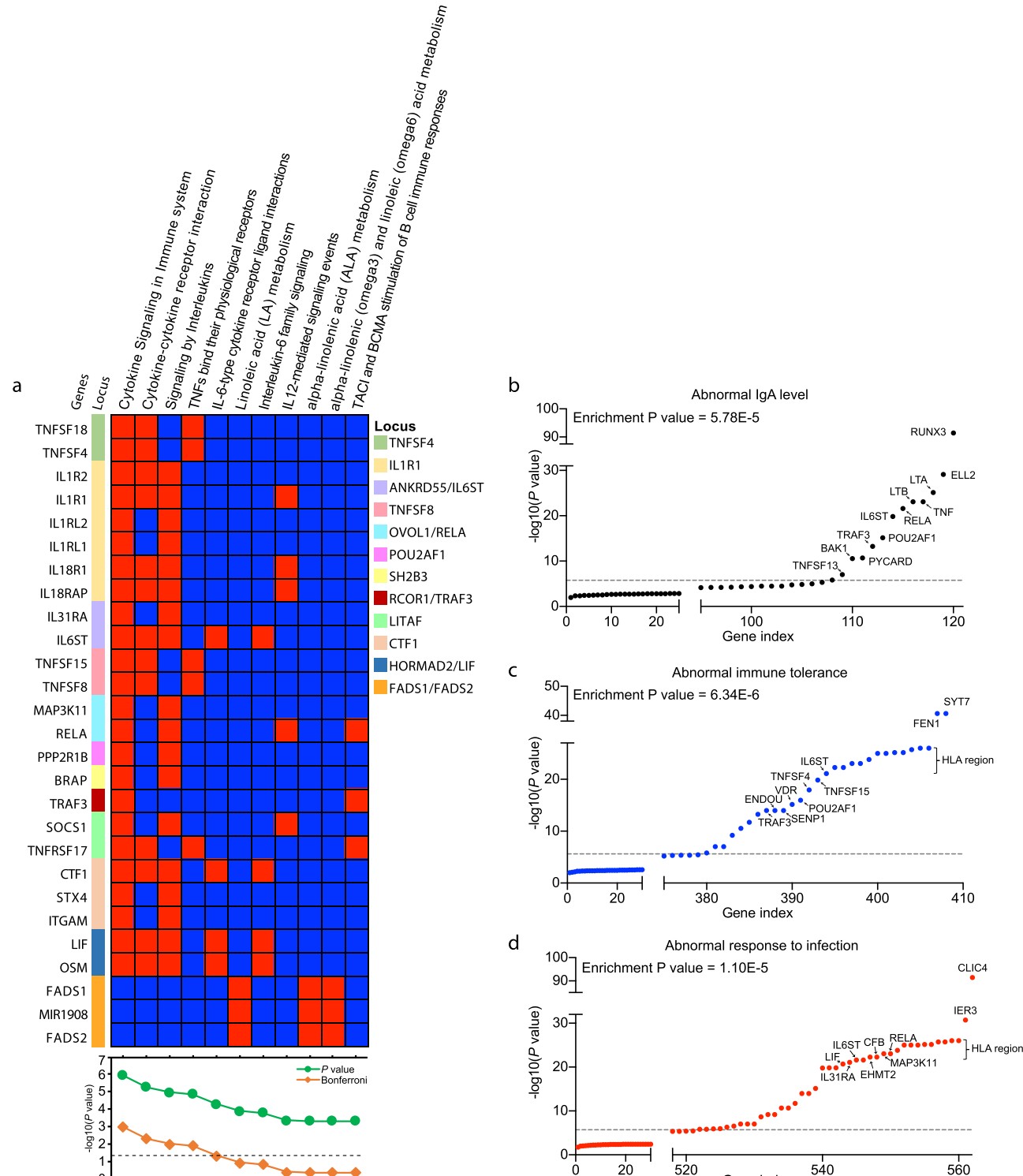

**Fig. 3 | Pathway and gene set enrichment analyses. a** Pathway enrichment analysis for genes at the significant GWAS loci (two-sided enrichment test *P*-values). **b** Gene-set enrichment for genes that cause abnormal IgA level. **c** abnormal immune tolerance; and (**d**) abnormal response to infection when genetically manipulated in mice. The *y*-axis shows the fixed effects meta-analysis −log10 (*P*-value) for the variant with the lowest two-sided unadjusted *P*-value in each candidate gene. The dashed line corresponds to genome-wide significance (*P* = 5 × $10^{-8}$). Enrichment *P*-value corresponds to the two-sided Fisher exact test comparing the observed number of genes with association signals below the genome-wide threshold against the number expected under binomial distribution.

the co-localized IgA level-associated loci as an exposure instrument and IgA nephropathy and tonsillectomy as disease outcomes. Using this strategy, we estimated significant causal effects between serum IgA levels and IgA nephropathy (inverse variance-weighted OR 9.70 per SD of exposure, 95%CI: 6.80–13.8, *P* < 0.001; Fig. 5c), supporting IgA level as a strong causal mediator of disease risk for these loci.

Sensitivity analysis confirmed that all co-localized loci contributed with concordant effects, there were no outlier effects, and there was no evidence of directional horizontal pleiotropy (Egger intercept test *P* = 0.58). Moreover, this effect remained highly significant when instrumental variables were expanded to encompass all genome-wide significant non-HLA loci for serum IgA levels (inverse variance-

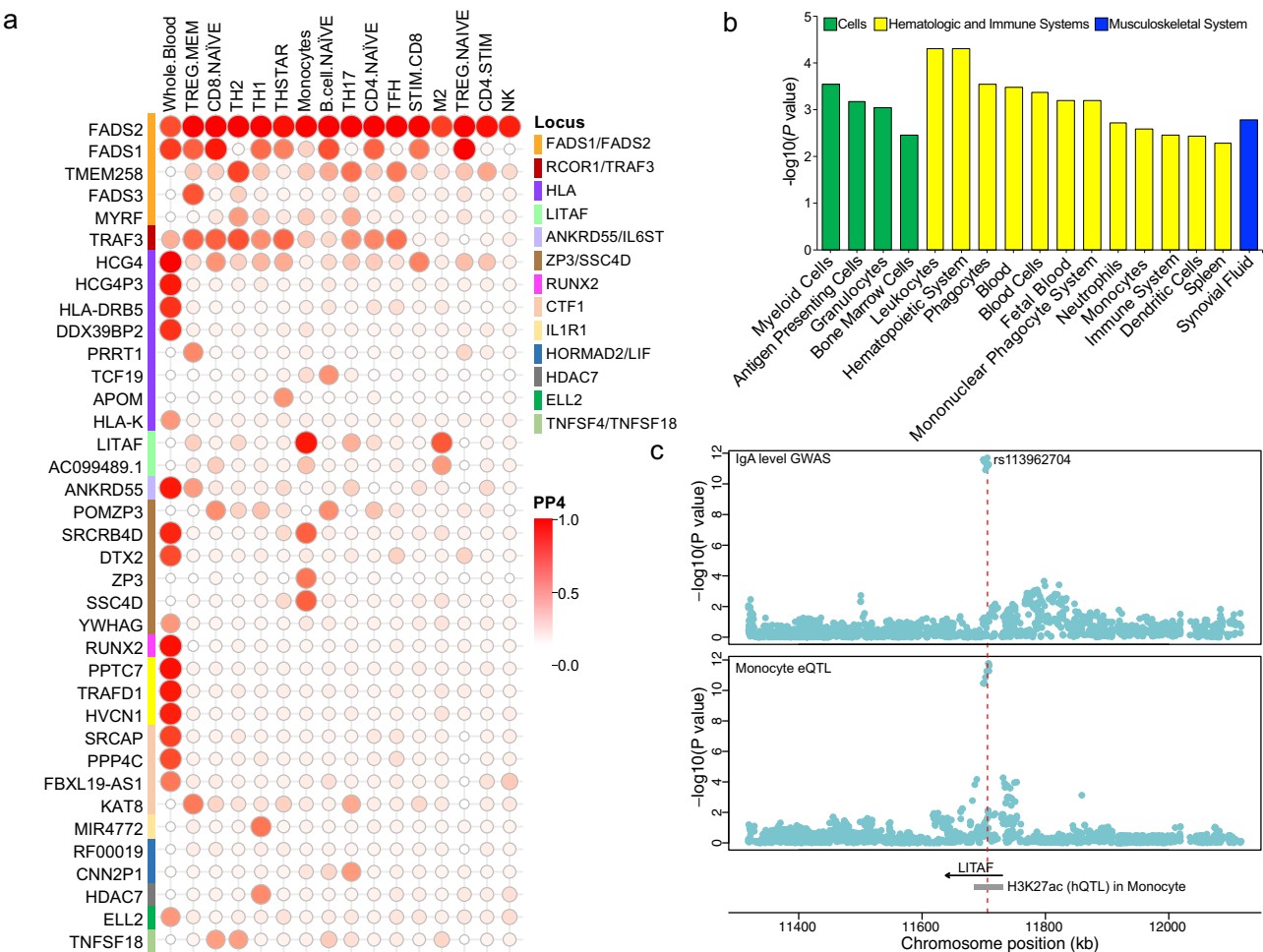

**Fig. 4 | Colocalization and enrichment analyses across different tissue/cell types. a** Colocalization analysis with gene expression QTLs (eQTLs) in whole blood and primary immune cells; each row indicates one gene and each column denotes one tissue/cell type; the color from red to white shows the probability of sharing the same causal variant between GWAS loci and eQTLs (PP4). **b** Tissue/cell type enrichment in DEPICT; the y-axis represents the −log10 of the two-sided adjusted p-value for the enrichment test and x-axis shows the first level MeSH tissue and cell type annotations. **c** Integrative analysis of eQTL and hQTL for *LITAF* locus. The upper and lower panels show the regional plots of the *LITAF* locus for IgA GWAS and eQTLs in monocyte, respectively. The y-axis represents the −log10 of the unadjusted two-sided p-value for the fixed effects GWAS meta-analysis (top) and the Wald test from linear regression of genotype on gene expression level (bottom); x-axis shows the chromosome positions. The lowest panel denotes the positions of *LITAF* gene and a hQTL peak (H3K27ac) in monocytes.

weighted OR = 2.49 per SD of IgA level, 95%CI: 1.56–3.99, $P < 0.001$). In contrast, we detected no significant causal effects between IgA levels and tonsillectomy ($P = 0.26$), or tonsillectomy and IgA nephropathy ($P = 0.96$).

## Genetic correlations with other complex traits studied by GWAS

We assessed genome-wide genetic correlations ($r_g$) of IgA levels with 52 complex traits and diseases, including 13 immune-mediated disorders, 23 infectious diseases, and 16 cardio-metabolic traits using stratified LD score regression[22,23] (Fig. 6a and Supplementary Table 12). In the analysis that excluded the HLA region (to remove potential bias from large effects and extended LD at this locus), we confirmed positive genetic correlation with IgA nephropathy ($r_g = 0.35$, $P = 0.002$), tonsillectomy ($r_g = 0.20$, $P = 0.01$), type 2 diabetes ($r_g = 0.18$, $P = 0.01$), and BMI ($r_g = 0.13$, $P = 0.03$). We observed a negative genetic correlation with Crohn's disease ($r_g = −0.19$, $P = 0.005$), celiac disease ($r_g = −0.21$, $P = 0.007$), and inflammatory bowel disease ($r_g = −0.13$, $P = 0.04$). The observed negative correlation with traits that involve gut inflammation could be potentially explained by the protective anti-inflammatory effects of mucosal IgA. Among infectious disease GWAS, we observed mainly negative genetic correlations, including with bacterial meningitis ($r_g = −0.47$, $P = 0.005$) and shingles ($r_g = −0.46$, $P = 0.009$), despite the fact

that most existing GWAS for infections are either underpowered or based only on self-report[61]. For most, but not all phenotypes, genetic correlations with and without HLA were comparable, as summarized in Supplementary Fig. 8 and Supplementary Table 12.

## Phenome-wide association studies (PheWAS)

To detect additional genetic associations, we derived a genome-wide polygenic score (GPS) for serum IgA levels and tested for its phenotypic associations using the meta-PheWAS approach across the UK Biobank and eMERGE-III datasets (but removing any participants included in the discovery GWAS, see Online Methods). In the combined analysis of 556,656 participants, we detected 31 significant phenotypic associations of the GPS (Fig. 6b and Supplementary Table 13). This included several protective associations with immune and inflammatory disorders, such as celiac disease (OR per SD = 0.54, $P = 4.6 \times 10^{-227}$), hypothyroidism (OR$_{SD}$ = 0.94, $P = 1.8 \times 10^{-19}$), type 1 diabetes (OR$_{SD}$ = 0.91, $P = 8.2 \times 10^{-12}$), and psoriasis (OR$_{SD}$ = 0.91, $P = 1.7 \times 10^{-10}$). Among significant risk associations were disorders of iron metabolism (OR$_{SD}$ = 1.31, $P = 4.0 \times 10^{-19}$) and hematuria, a common manifestation of IgAN (OR$_{SD}$ = 1.04, $P = 1.0 \times 10^{-8}$). To assess which of these associations were driven by the HLA region, we next repeated meta-PheWAS after excluding all HLA variants from the GPS (Fig. 6c and Supplementary

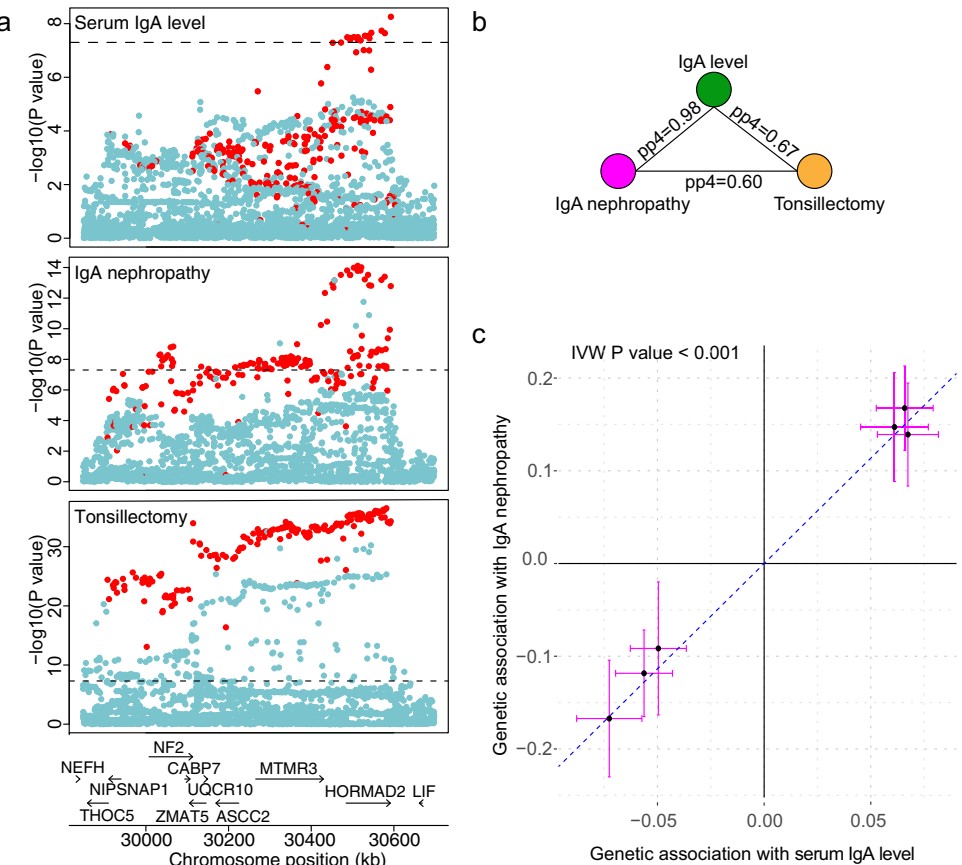

**Fig. 5 | Colocalization and Mendelian randomization analyses based on GWAS for serum IgA levels, IgA nephropathy and tonsillectomy. a** Regional plot of the *HORMAD2/LIF* locus for IgA levels (without the deCODE-Lund cohort, top panel), IgA nephropathy (middle panel), and tonsillectomy (lower panel). The two-sided unadjusted *P*-value corresponds to the fixed effects GWAS meta-analysis. **b** Co-localization analysis of *HORMAD2/LIF* locus across the three traits; PP4 is the posterior probability of co-localization. **c** Mendelian randomization analysis using

IgA level (GWAS *N* = 41,263) as an exposure, IgA nephropathy (GWAS *N* = 38,897) as an outcome, and co-localizing loci as instruments. The error bars correspond to 95% confidence intervals for the effect size. The *x* and *y* axis represent effect sizes of the genetic variants associated with the exposure and outcome, respectively. *P*-value corresponds to the two-sided inverse variance weighted (IVW) Mendelian Randomization test.

Table 13). In this analysis, we detected only two phenome-wide-significant associations: a protective association with Celiac disease (OR per SD = 0.86, $P$ = 4.6 × $10^{-12}$) and a risk association with morbid obesity ($OR_{SD}$ = 1.09, $P$ = 3.0 × $10^{-5}$). These associations were direction-consistent with our genome-wide genetic correlation analyses.

Given these results, we next applied Mendelian randomization approach to resolve potential causal relationships between serum IgA levels, Celiac disease, and BMI. For instrumental variables, we used independent genome-wide significant alleles from this study (excluding the HLA region), and from the largest studies for Celiac disease[62] and BMI[63]. Interestingly, we detected no significant causal effects between serum IgA levels and Celiac disease or BMI. In reverse causality analyses we observed no causal effect of Celiac disease on serum IgA levels, however, there was a highly significant causal effect of BMI on serum IgA levels (inverse variance weighted effect = 0.12, 95%CI: 0.05–0.19, $P$ < 0.001). In sensitivity analyses, there was evidence of directional pleiotropy (Egger intercept test $P$ = 0.004), but the causal effect of BMI became stronger and more significant when the balanced pleiotropy assumption was relaxed (Egger regression effect = 0.38, 95%CI: 0.19–0.58, $P$ < 0.001). These results suggest that elevated serum IgA levels in individuals with metabolic syndrome may represent a consequence rather than a cause of obesity.

## Discussion

Our study provides multiple insights into the genetic regulation of serum IgA levels and dissects shared genetic effects between IgA levels and several human diseases. Our large multi-ethnic meta-analysis identified 20 genome-wide significant loci, 11 of which were novel. These loci encode genes enriched in immune-related pathways, with 13 candidate genes demonstrating IgA abnormalities when genetically manipulated in mice. The complementary enrichment analyses based on gene expression across multiple tissues/cell types highlighted primary immune cells, mainly T cells, B cells and monocytes/macrophages, as the most likely effector cell types regulating IgA production.

Previous smaller GWAS for serum IgA levels have been limited to European or East Asian ancestry, while African and other ancestries have not been included in these studies. Our multi-ethnic cohort included four diverse ancestral groups and allowed us to demonstrate that African ancestry was associated with higher serum IgA levels compared to other ancestries. Consistent with this observation, IgA-increasing alleles were more frequent in African ancestry compared to non-African participants, and we demonstrated that the African ancestry populations had the highest GPS compared to all other reference populations. Positive correlation of effect sizes with AFR-EUR differences in allele frequency for the top 1% IgA-increasing alleles is supportive of polygenic adaptation. This correlation was not significant when extended genome-wide, but the power is limited by the sample size of our GWAS, low SNP-based heritability of serum IgA levels, and incomplete knowledge about the true underlying genetic architecture. Although not statistically significant in genome-wide analysis, the negative correlation coefficient between IgA-increasing effects and European tSDS scores is consistent with the hypothesis

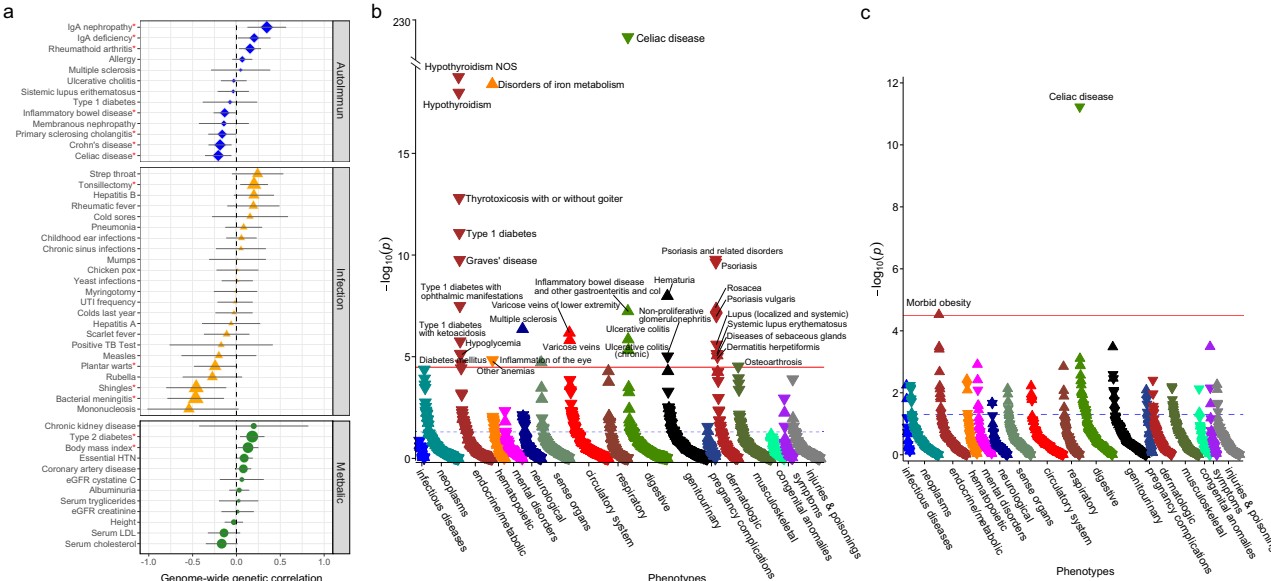

**Fig. 6 | Genetic relationships between IgA levels and human disease traits.**
**a** genome-wide genetic correlation analyses between IgA levels and autoimmune, infectious, and cardio-metabolic traits after exclusion of the HLA region (*$P < 0.05$; two-sided unadjusted $P$-values for genetic correlation by LD score regression). Supplementary Table 12 provides genetic correlations with and without HLA with $P$-values for each trait, and references to the original GWAS studies; the error bars correspond to 95% confidence intervals for genetic correlation coefficients. **b** Meta-PheWAS of genome-wide polygenic score (GPS) for IgA levels across eMERGE-III and UKBB biobanks (total $N = 556,656$). **c** Meta-PheWAS of GPS for IgA levels without the HLA region (UKBB and eMERGE-III, total $N = 556,656$). In (**b**) and (**c**), the $y$-axis shows −log10 ($P$-value); each triangle represents an individual phenotype

(phecode) tested as an outcome against the GPS for IgA levels as a predictor; an upward triangle indicates a positive (risk) association, while a downward triangle indicates a negative (protective) association; two-sided unadjusted $P$-value corresponds to the fixed effects meta-analysis across the two biobanks based on logistic regression adjusted for age, sex, site, genotyping batch, and principal components of ancestry; the red line corresponds to the Bonferroni-corrected significance threshold for 1523 phecodes tested (alpha = 0.05/1523 = $3.28 \times 10^{-5}$); the phenotypes are grouped by organ system (or relevant disease category) and sorted based on their statistical significance within each group. Supplementary Table 13 provides a comparison of significant PheWAS associations with and without the HLA region.

that recent selection might have favored IgA-lowering alleles in Europeans. However, environmental exposures correlated with African ancestry may also be contributing to the observed population differences in IgA levels. Our study was not designed to identify environmental effects, and the existing methods for detecting multi-locus selection have low power and lack sensitivity[24].

Our systematic co-localization analysis of GWAS associations and eQTLs in whole blood and 13 different immune cell types prioritized new biologically plausible candidate genes and target cell types for 14 of the 20 significant loci, providing an extensive resource for follow up studies. We also observed that many eQTL co-localizations were present only in specific immune cell subtypes. As an example, variants within the *LITAF* locus co-localize specifically with *LITAF* eQTL but only in monocytes, pointing to monocyte lineage as the most likely causal cell type for this locus. In contrast, *TRAF3, HDAC7, IL1R1* and *ANKRD55* loci co-localize with eQTLs specifically in human T cell lineages, prioritizing T cells for functional studies of these loci.

In the analysis of genetic relationships with other traits, we observed shared polygenic determination between IgA levels and several immune, infections, renal, and cardio-metabolic traits. Our findings support the protective role of IgA in susceptibility to various infectious pathogens as well as inflammatory bowel disease. At the same time, we observed a positive genetic correlation between IgA levels and IgA nephropathy, a common form of kidney disease due to IgA deposition. In particular, Mendelian randomization analysis suggested a causal role for elevated IgA levels in the pathogenesis of IgA nephropathy. Ongoing IgA nephropathy clinical trials targeting pathways that reduce IgA levels will shed light on this hypothesis[64].

Our study has several important limitations. First, genome-wide summary statistics from the largest previously published study by deCODE-Lund[20] were not publicly available, thus we were limited to the use of SNPs with $P < 1.0 \times 10^{-6}$ from that study in our meta-analysis.

This reduced our power for new discovery and produced uneven genomic coverage in the meta-analysis. Consequently, our cis-eQTL co-localization analyses were restricted to non-deCODE-Lund cohorts and were additionally limited by low power of presently available eQTL datasets for primary immune cells. Similarly, genome-wide genetic correlation analyses were driven largely by our non-deCODE-Lund cohorts. These analyses were additionally limited by sample sizes of previously published GWAS, and the fact that infectious traits have been generally understudied by GWAS. Our GPS modeling strategy relied on 1000 Genomes for LD reference, since more diverse panels, such as TopMed, are not publicly available, precluding their use for this purpose. The key limitations of our PheWAS approach included missingness of "real life" EHR data, inadequate ICD coding for some relevant conditions (e.g., IgA nephropathy), and the fact that rare diagnoses (e.g., IgA deficiency) are generally poorly captured in population-based biobanks.

In summary, we reported 20 genome-wide significant loci associated with serum IgA levels, and we prioritized potential effector genes and cell types for 70% of the loci. We demonstrated that IgA levels are positively correlated with African ancestry. We further characterized shared genetic architecture between serum IgA levels and other complex traits, demonstrating that while IgA-increasing alleles appear to have protective effects against infections, they may represent risk factors for selected auto-immune, kidney, and cardio-metabolic diseases. Notably, the risk of IgA nephropathy appears to be causally related to elevated IgA levels with several genome-wide significant loci co-localizing between the two traits.

## Methods
### Measurement of serum IgA levels
Serum total IgA levels were measured via a previously optimized and validated 'sandwich' ELISA protocol. 96-well plates were coated with an

IgA capture antibody (AffiniPure F(ab')$_2$ Fragment of Goat Anti-Human Serum IgA, α Chain Specific; Jackson Immuno Research, #09-006-011), covered, and let to sit on a shaker for 60 min. The plate was subsequently washed three times with a TBS Wash buffer. Next, 250-μL Blocking Buffer was added to the wells, and the plate was again covered, and set on the shaker for 90 min. After aspirating the wells of the blocking buffer, 100 μL/well of samples and standards were added, all in duplicate. Standards were diluted 1:3 in series beginning with 500 ng/mL. The plate was left to incubate for 120 min, after which samples were aspirated and the plate was washed thoroughly five times. Next, 100 μL of secondary/detection antibody (Mouse monoclonal Ab to Human IgA horseradish peroxidase conjugate, Abcam Bio #ab7383), was added at 62.5 ng/mL to the wells and left to incubate for 60 min. The plate was washed an additional five times. Next, 100 μL of TMB 2 component ELISA substrate (KPL, 50-76-00) was added to the wells, and left to incubate for 12 min. 100 μL of 2 M H2SO4 (Sigma Aldrich, #339741) was added to each well at the end of the 12 min to stop the reaction, per substrate manufacturer recommendation. The samples were then measured using the BIO-TEKPowerWaveTM XS and KC-Jr. plate reading software. The plate was read at 450 nm, with a 630 nm reference wavelength, per substrate manufacturer recommendation. Samples with an internal CV greater than 10% were re-run. Samples that fall outside of the standard range (500–0.2 ng/mL) were re-run at appropriate dilutions.

### MESA cohorts

The Multi-Ethnic Study of Atherosclerosis (MESA) is a diverse population-based cohort of participants recruited prospectively for studies of cardiovascular disease[65]. The recruitment took place across six clinical centers in the United States. The participants were genotyped using the Affymetrix Human SNP array 6.0[66]. We measured serum IgA levels using standardized ELISA protocol (see above) in $N = 5420$ participants, and these individuals were included in the GWAS analysis. The standard genotype quality control (QC) filters included per-SNP genotyping rate >95%, per-individual genotyping rate >90%, MAF > 0.01, and HWE test $p$-value $>1 \times 10^{-5}$ within each ancestry group. We assessed for cryptic relatedness and duplicates, and we excluded one individual from any pairs with estimated pairwise kinship coefficients >0.05. Gender of each individual was imputed based on the analysis of sex chromosome markers and individuals with gender mismatch against records were excluded. The imputation analysis was carried out using Minimac3[67] after pre-phasing in Eagle V2.3[68] and using 1000 Genomes (Phase 3) as ref. [69]. A total of 11,102,943 common high-quality markers ($R^2 > 0.8$ and MAF > 0.01) were imputed and used in the downstream GWAS analyses. The principal component analysis (PCA)[70] was used to assign genetic ancestry to the participants based on co-clustering with the major continental reference populations of the 1000 Genomes Phase 3[69]. After exclusions of ancestry outliers, we identified 5 distinct ancestral clusters (European: $N = 2280$, African: $N = 1275$, Admixed 1: $N = 474$ and Admixed 2: $N = 750$, and East Asian: $N = 641$). Subsequent GWAS analyses were then performed within each ancestry cluster. For ancestry adjustment, we re-run PCA within each cluster, defined significant PCs by Tracy-Widom test, and included the significant PCs for each cluster as covariates in the association testing.

### Electronic medical records and genomics (eMERGE-III) cohorts

The eMERGE consortium consists of 12 medical centers with electronic health records (EHRs) linked to genome-wide genotype data for 102,138 individuals. The serum IgA level was extracted by a lab value query performed by eight active sites in the eMERGE phase III. The genotyping and imputation of the eMERGE cohort have been previously described in detail[71–73]. Briefly, we implemented the mimimac3 imputation model with HRC1.1 as references for each genotyping platform in a separate batch. After imputation, we merged all the 81

imputed batches based on genomic position. Post-imputation marker QC filters included MAF $\geq 0.01$ and $R^2 \geq 0.8$ in $\geq$75% of 81 imputation batches. We also excluded duplicates and cryptic relatedness in the given cohort determined by the estimated pairwise kinship coefficients >0.05. The genetic ancestry for each individual was assigned based on PCA with reference populations of the 1000 Genomes Phase 3[69]. After exclusions of ancestry outliers, the final dataset with matched serum IgA phenotypes consisted of 6047 individuals across five ancestral groups: $N = 4261$ European ancestry individuals (6,625,959 high-quality imputed markers), $N = 476$ African ancestry individuals (7,006,144 high-quality imputed markers), $N = 73$ East Asian ancestry individuals (5,410,844 high-quality imputed markers), $N = 235$ Admixed 1 cohort (7,106,891 high-quality imputed markers) and $N = 1002$ Admixed 2 cohort (7,296,903 high-quality imputed markers). Like the MESA cohort analysis, we repeated the PCA within each ancestry cohort and defined significant PCs to be used as covariates in association testing.

### German, French, Chinese, Japanese, and U.S. cohorts

The description of these cohorts including recruitment and measurements of serum IgA levels have been published previously[74]. For this study, we included only individuals who were ascertained as 'healthy population controls' for GWAS studies, thus these cohorts were not enriched in any specific disease type. The genotyping was performed using Illumina 550v3 (US cohort), Illumina 370-Duo (French cohort), Illumina MEGA v1.0 (Chinese and Japanese cohorts) and Illumina MEGA v1.1 (German cohort). The standard genotype QC filters included per-SNP genotyping rate >95%, per-individual genotyping rate >90%, MAF >0.01, and HWE test $p$-value $>1 \times 10^{-5}$ within each cohort. We assessed for cryptic relatedness and duplicates, and we excluded one individual from any pairs with estimated pairwise kinship coefficients >0.05. Gender of each individual was imputed based on the analysis of sex chromosome markers and individuals with gender mismatch against records were excluded. The imputation was performed using MACH 1.0 for pre-phasing and then Minimac3 for imputation based on ancestry-matched reference panels of 1000 Genome Project (Phase 3). We performed PCA of each dataset to exclude outliers and define the number of significant PCs by Tracy-Widom test. The final numbers of individuals and high-quality markers ($R^2 > 0.8$, MAF>0.01) used in downstream GWAS analyses were as follows: the German cohort of $N = 156$ healthy individuals (7,612,078 markers); the French cohort of $N = 103$ healthy individuals (7,096,980 markers), the Chinese cohort of $N = 467$ healthy individuals (5,113,877 markers), the Japanese cohort of $N = 776$ healthy individuals (6,673,613 markers); and the U.S. cohort of $N = 93$ healthy individuals (7,439,363 markers).

### Swedish cohort

We obtained summary statistics for GWAS for total IgA levels for 9,617 participants; the ascertainment, genotyping, and analysis of this cohorts has been published previously[16]. In order to improve marker density, we re-imputed this cohort based on association estimates of the genotyped markers from the summary statistics using ImPG V1.0[75] software and the 1000 Genomes (Phase 3) European reference[69]. Using ImPG software, we derived the posterior mean of z-scores at untyped SNPs given the z-scores at typed SNPs and the correlation matrix among all pairs of SNPs induced by their linkage disequilibrium (LD) that were estimated using the reference panel. The effect size and standard error for each imputed SNP were then estimated based on its imputed z-score and reference allelic frequency as described previously[76]. A total of 6,907,390 variants were imputed with high quality ($R^2 > 0.8$ and MAF > 0.01) and included in our downstream analysis.

### The deCODE-Lund cohort

The analysis of this cohort composed of 16,883 participants from Iceland and 2151 individuals from southern Sweden has been published

previously[20]. The association summary statistics for IgA levels including effect size, P-value and minor allele frequencies for 4699 variants with $P < 1 \times 10^{-6}$ were provided by the authors in their publication[20]. In order to incorporate this dataset using fixed-effects meta-analysis, the unbiased standard error for each variant was derived using the following equation[77]:

$$SE = \sqrt{\frac{1 - 2p(1-p)b^2}{2p(1-p)n}}$$

where $b$ is the standardized allelic effect on IgA levels, $p$ is the minor allele frequency, and $n$ is the sample size.

## Genome-wide association studies and multi-ancestry meta-analysis

We conducted a multi-ancestry meta-analysis of 16 cohorts with genome-wide data for 22,229 diverse study participants with 4699 suggestive association signals ($P < 10^{-6}$) from the previous GWAS by deCODE Genetics and Lund University (19,034 participants of North European ancestry)[20]. Multi-ancestry cohorts were classified into ancestry-specific strata based on global principal component analysis. In each sub-cohort, serum IgA levels were log-transformed and expressed as standard-normalized residuals from regression of log-transformed IgA levels against age and sex. We performed genome-wide association testing in each cohort for the markers that were imputed at high quality ($r^2 > 0.8$) using a linear regression model under additive coding of the dosage genotypes, and with adjustment for cohort-specific significant principal components (PCs) of ancestry[78]. To quantify potential inflation of type I error due to stratification or technical artifacts, we estimated the genomic inflation factor for each cohort but detected no substantial inflation with lambda <1.05 in each individual study. We performed a fixed-effects as well as TransMeta random effects meta-analysis to combine the results of all 17 individual cohort summary statistics using METAL[79] and TransMeta[21] software, respectively. All significant loci were further assessed for heterogeneity by derivation of Heterogeneity Index ($I^2$) and by testing using Cochrane's heterogeneity test in PLINK[80]. The quantile-quantile plot of the final meta-analysis showed no global departures from the expected null distribution, with the genomic inflation factor estimated at 1.016 (Supplementary Fig. 2). The genome-wide significant signals were defined by the generally accepted $P < 5.0 \times 10^{-8}$ and signals with $P < 1.0 \times 10^{-6}$ were considered as suggestive.

## Conditional analyses

To detect independent variant associations at each genome-wide significant locus, we performed stepwise conditional analysis within each cohort, followed by meta-analysis of the conditioned summary statistics. For the deCODE-Lund and Swedish cohorts, only summary statistics were available, thus we used the approximate conditioning using conditional & joint association analysis method (COJO, GCTA software[81]) with 1000 Genomes Phase 3[69] (European populations only) for LD reference. All other cohorts had genome-wide genotype data available, and we used primary genotype data to perform conditioning before meta-analysis.

## Tissue/cell type and pathway enrichment analyses

The region of each GWAS locus was defined by first selecting all proxy SNPs in LD ($r^2 > 0.5$) with the lead SNP, then extending the genomic region 250 kb upstream and downstream of the first and last proxy SNP based on genomic position. Each region was then annotated using Ensembl human gene annotations. Gene sets were created for all genome-wide significant regions but excluding the HLA region. For tissue/cell type enrichment, we used DEPICT (Data-driven Expression-Prioritized Integration for Complex Traits) to test for tissues and cell-types in which genes from the associated regions were highly

expressed as previously described[26]. Next, for each tissue, empirical enrichment p-values were computed by repeatedly sampling random sets of loci from the entire genome to estimate the null distribution for the enrichment statistic. For pathway enrichment analysis, we used established pathways from the databases including Molecular Signatures Database (MSigDB C2), KEGG, BioCyC, REACTOME, Pathway Interaction Database. Statistical significance for enrichment was set at FDR q-value <0.05.

## Functional annotations

We first defined each GWAS locus by ±400 kb of the genome-wide significant index SNP. We annotated all transcripts within these intervals using the latest assembly of the human genome (hg19) to create sets of positional candidate genes for each locus. Using ANNOVAR software[82], we annotated all variants within the region that were in LD ($r^2 > 0.5$) with the top SNP, including all known coding, splicing, and 3′ UTR and 5′UTR variants. To prioritize the candidate genes at each of the GWAS loci, we next performed colocalization analyses based on our meta-analysis statistics and gene expression QTLs in whole blood quantified from 31,684 individuals[30], as well as 13 human immune cell types from the Database of Immune Cell eQTLs (DICE) project[31]. After harmonization of effect alleles, we identified all co-localized eQTLs mapping to the region of the index SNP ± 400 kb using Coloc Package in R[83]. Co-localization with PP4 greater than 0.7 was considered as strong evidence in support of shared causal SNPs, while PP4>0.5 was considered as suggestive. To test for histone QTL effects in immune cells, we interrogated all GWAS index SNPs and their proxies ($r^2 > 0.5$) against histone QTL of three major human immune cell types (CD14+ monocytes, CD16+ neutrophils, and naive CD4+ T-cells) based on the analysis of ~200 individuals[40].

## Protein–protein interactions (PPI) network analyses

Protein–protein interactions among the positional candidate genes at the GWAS loci were predicted using InWeb_InBioMap[84], a curated and computationally derived regulatory network of 420,000 interactions (Supplementary Fig. 6). We used high confidence interactions defined by the recommended cut off confidence score <0.1. The candidate gene network contained a total of 48 genes and 52 interactions. The network components were grouped into ten modules based on their connectivity. Functional and pathway enrichments within each module were identified based on Gene Ontology, KEGG, and Reactome databases. We used ToppGene Suite[85] to calculate interaction enrichment p-values for each gene; a Bonferroni-corrected $P < 0.05$ was used as enrichment significance cut-off.

## Intersection with related mouse phenotypes

We evaluated genes that when genetically manipulated cause abnormal immune phenotypes in mice based on the comprehensive MGI phenotype ontology database. The following mouse phenotypes were evaluated: abnormal IgA levels (MP:0020171); abnormal immune tolerance (MP:0005000); and abnormal response to infection (MP:0005025). The human orthologs of these genes were obtained with the Human–Mouse Disease Connection web tool (http://www.informatics.jax.org/humanDisease.html). The significance of intersections between these gene sets and the list of positional candidate genes from GWAS was determined using a hypergeometric test.

## Heritability and genetic correlations with other phenotypes

SNP-based heritability of IgA level GWAS was estimated by LD score regression model using the LDSC software[22]. The LD score for each SNP was estimated based on LD matrices derived from ancestry-matched 1000 Genome Project Phase 3 populations. To investigate the shared genetic architecture between IgA levels and other phenotypes, we first collected the summary statistics of autoimmune and inflammatory disorders and cardio-metabolic traits from the LD-hub[86] or GWAS

catalog[27], and summary statistics for infection-related phenotypes were provide by 23andMe[61]. For each phenotype, we used GWAS summary statistics from the largest GWAS available with a minimum coverage of 2 million SNPs. We excluded traits with estimated SNP-based heritability <1%. The genetic correlations were next estimated using bivariate LD score regression[22] with ancestry-matched 1000 Genome Project Phase 3 as the LD reference panel.

## Pleiotropic annotation of GWAS loci

To systematically cross-annotate our loci against all previously published GWAS findings, we downloaded the GWAS catalog[27]. We filtered all published SNPs that were associated with any disease phenotype or trait at a genome-wide significance ($P < 5 \times 10^{-8}$) and resided within the genomic regions of association with IgA level. For each SNP associations, we manually verified the direction of effect for a reference allele based on original GWAS publications. Next, each selected SNP from the catalogue was queried against our GWAS results to extract the odds ratios and $P$-values for associations with IgA levels. The directionality of allelic effects was assessed to identify pleiotropic alleles with concordant or opposed effects. We calculated a maximum $r^2$ between SNPs associated with each cataloged trait and the independent SNPs from our study based on 1000 Genomes Project Phase 3 data[87]. We defined shared susceptibility alleles if $r^2$ between the top SNPs exceeded 0.5. We constructed a susceptibility overlap map that connected each of the IgA GWAS loci to the previously associated GWAS traits and highlights associations with SNPs in high LD with the top signals. The map was visualized with Cytoscape v.3.6 software[88]. Moreover, given a large number of overlapping GWAS loci between IgA levels, IgAN, and tonsillectomy, we performed systematic locus colocalization analyses based on regional summary statistics using Coloc software[83].

## Genome-wide polygenic score and tests of polygenic adaptation

For the purpose of testing cumulative effects of genetic determinants of serum IgA levels, we derived a genome-wide polygenic score (GPS) for IgA levels using LDPred algorithm[89] which estimates posterior mean causal effect sizes from GWAS summary statistics by assuming a prior for the genetic architecture and LD information from a reference panel. We used GWAS summary statistics for IgA levels assuming 1% fraction of causal variants and for LD reference we used 1000 Genomes (Phase 3), all populations except South Asian ancestry that was not represented in our cohorts[69]. To test for population differences in the GPS distributions, we applied the score to the 1000 Genomes participants and performed pairwise t-test comparisons between the reference populations. To test for potential polygenic adaptation, we performed rank correlation analysis of LDPred posterior mean causal effect sizes versus AFR-EUR allelic frequency difference calculated based on 1000 Genomes reference populations; given our LDPred priors, we used only the top 1% of GPS variants with the highest weights for this analysis (the weights for the remaining variants were close to zero as expected). Next, we extended this analysis genome-wide by testing for genetic correlation between AFR-EUR allelic frequency difference and GWAS effect sizes for IgA-increasing alleles using LD score regression[23]. In this analysis, we replace GWAS Z-scores for one of the two traits with a Z-score-normalized IgA-increasing allele frequency difference. Lastly, we used singleton density score (SDS) method based on contemporary European genome sequences that infers recent (~2000–3000 years) changes in derived allele frequencies[24]. We derived trait-SDS (tSDS) scores from the raw European SDS scores by polarizing them to IgA-increasing alleles (we reset the sign of SDS scores such that positive values indicate increased frequency of the IgA-increasing allele instead of the derived allele). The tSDS scores were standardized to mean 0 and variance 1 within each 1% allelic frequency bin and tested for genome-wide genetic correlation with IgA-increasing effects using

LD score regression[23]. These analyses were performed with R version 3.4 (CRAN).

## Meta-phenome-wide association study (Meta-PheWAS)

We performed a meta-PheWAS analysis for the GPS across the UK Biobank (UKBB, $N = 460,364$ participants) and Electronic Medical Records and Genomics-III (eMERGE-III, $N = 96,292$, after excluding those participants analyzed in the GWAS for IgA levels). The eMERGE-III genotype data was processed in the same way as for GWAS described above, but for ancestry adjustments we performed principal component analysis of the entire eMERGE-III cohort using FlashPCA[90] on a set of 48,509 common (MAF>0.01) and independent variants (pruned in PLINK with–indep-pairwise 500 50 0.05 command). The first 3 PCs were included as a covariate in PheWAS. The UKBB is a large prospective population-based cohort that enrolled individuals ages 40–69 for genetic studies[91]. All 488,377 UKBB participants underwent genotyping with Affymetrix's UK Biobank Axiom and UK BiLEVE Axiom arrays with genotype imputation using a 1000 Genomes reference panel with IMPUTE4 software[92–94]. We applied similar QC filters to eMERGE-III, retaining 9,233,643 common (MAF ≥ 0.01) variants imputed with high confidence ($R^2 \geq 0.8$). For principal component analysis using FlashPCA[90], we used a set of 35,226 variants with MAF > 0.01 and pruned using–indep-pairwise 500 50 0.05 command in PLINK. The first 3 PCs were used as covariates in PheWAS. To harmonize coded diagnoses between UKBB and eMERGE-III we converted all available ICD-10 codes to ICD-9-CM system given that the great majority eMERGE-III diagnoses were coded using ICD-9-CM. After the conversion, eMERGE participants had a total of 20,783 ICD codes that were then mapped to 1817 distinct phecodes. The 488,377 UKBB participants had a total of 10,221 ICD codes mapped to 1523 phecodes. Phenome-wide associations were then performed using the PheWAS R package[95]. The case definition required a minimum of two ICD-9 codes from the "case" grouping of each phecode, while "control" group had no ICD-9 codes relevant to the tested phecode. In total, 1,523 overlapping phecodes were tested in both UKBB and eMERGE-III using logistic regression after adjusting each analysis for age, sex, study site, genotyping batch, and 3 PCs of ancestry. The meta-PheWAS across both datasets was performed using metagen under fixed effects model in PheWAS R library[95]. To establish significant disease associations in PheWAS, we set the Bonferroni-corrected statistical significance threshold at $3.28 \times 10^{-5}$ (0.05/1523) correcting for 1523 independent phecodes tested.

## Mendelian randomization analyses

Two sample mendelian randomization (MR) analyses were performed using genetic variants as instruments to test the causal effects between an exposure and an outcome. Only SNPs independently associated with the exposure at a genome-wide significance ($P < 5 \times 10^{-8}$) were used as instruments in MR studies. We excluded HLA alleles from all instruments, since these alleles are likely to exhibit strong pleiotropic associations with a wide range of immune-related outcomes. For primary hypothesis testing, we used inverse variance weighted MR model under the assumption of balanced pleiotropy by meta-analyzing SNP specific Wald estimates using multiplicative random effects. The random effects model was chosen to account for any potential heterogeneity. Given a total of eight bi-directional MR tests performed between IgA levels and IgAN, tonsillectomy, Celiac disease, and BMI, we used a Bonferroni-corrected significance threshold alpha = 0.05/8 = 0.00625. We additionally tested for the presence of directed horizontal pleiotropy using Egger test for non-zero intercept. Additional sensitivity analyses were performed by testing for outlier effects, and relaxing the assumption of balanced horizontal pleiotropy by using median-based estimator, mode-based estimator, and Egger regression methods. The testing was conducted using the TwoSampleMR package[96].

## Ethics statement

All subjects provided informed consent to participate in genetic studies, and the Institutional Review Board of Columbia University approved our studies under the following protocol numbers: IRB-AAAC7385 (primary analysis), IRB-AAAQ9205 (eMERGE-III analysis), IRB-AAAC9458 (MESA SHARe analysis), and IRB-AAAS3500 (UK Biobank analysis).

## Reporting summary

Further information on research design is available in the Nature Portfolio Reporting Summary linked to this article.

## Data availability

The MESA SHARe genotype and phenotype data (including serum IgA levels measured in this study) are available through dbGAP, accession number phs000209.v13.p3. The Electronic Medical Records and Genomics-III (eMERGE-III) imputed genotype and phenotype data are available through dbGAP, accession number: phs001584.v2.p2. The UK Biobank genotype and phenotype data are available through the UK Biobank web portal. Genotype data for other cohorts are available through dbGAP, accession number: phs000431.v3.p1. The 1000 Genomes data are available publicly through https://www.internationalgenome.org/category/data-access/. GWAS summary statistics are available for download from https://www.columbiamedicine.org/divisions/kiryluk/study_gwas_stat_IgA.php.

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

## Acknowledgements

The authors are grateful to all study participants for contributing DNA and serum samples for the purpose of our genetic studies. We would also like to acknowledge Dr. Andre Franke, Institute of Clinical Molecular Biology, Christian Albrechts University of Kiel, Kiel, Germany and Dr. Alexandra Zhernakova, Department of Genetics, University of Groningen, University Medical Center Groningen, Groningen, the Netherlands for providing summary statistics for the intestinal microbiome GWAS. This work was funded by the National Institute of Diabetes and Digestive and Kidney Diseases (NIDDK) grant numbers R01-DK105124 (K.K.), RC2-DK116690 (K.K.), R01-DK078244 (J.N., B.A.J.), and R01-DK082753 (A.G.G., J.N., K.K., B.A.J.) with additional support by R01-LM013061 (K.K., C.W.), R01-LM006910 (G.H.), U01-HG008680 (K.K.), U01-AI152960 (K.K.), K25-DK128563 (A.K.), UL1-TR001873 (A.K., K.K.), K01-DK106341 (C.R.), and R03-DK122194 (C.R.). J.F. and F.E. were funded by the Deutsche Forschungsgemeinschaft (DFG, German Research Foundation)—CRU 5011—Project-ID 445703531. T.R. is funded by R01-MD012467, R01-NS029993, R01-NS040807, U24-NS107267, U19-AG065169, UL1-TR002736, KL2-TR002737, and the Florida Department of Health. MESA and the MESA SHARe project are conducted and supported by the National Heart, Lung, and Blood Institute (NHLBI) in collaboration with MESA investigators. Support for MESA is provided by contracts 75N92020D00001, HHSN268201500003I, N01-HC-95159, N01-HC-95160, N01-HC-95161, 75N92020D00005, 75N92020D00002, 75N92020D00003, N01-HC-95162, 75N92020D00006, N01-HC-95163, 75N92020D00004, N01-HC-95164, 75N92020D00007, N01-HC-95165, N01-HC-95166, N01-HC-95167, N01-HC-95168, N01-HC-95169, UL1-TR-000040, UL1-TR-001079, UL1-TR-001420, UL1-TR-001881, and DK063491. Funding for SHARe genotyping was provided by NHLBI Contract N02-HL-64278. Genotyping was performed at Affymetrix (Santa Clara, CA, USA) and the Broad Institute of Harvard and MIT (Boston, MA, USA) using the Affymetrix Genome-Wide Human SNP Array 6.0. Parts of this study have been conducted using the UKBB Resource under UKBB project ID number 41849. The eMERGE network is funded by the National Human Genome Research Institute (NHGRI) through the following grants: U01HG008657 (Group Health Cooperative/University of Washington); U01HG008680 (Columbia University Health Sciences); U01HG008685 (Mass General Brigham); U01HG008672 (Vanderbilt University Medical Center); U01HG008666 (Cincinnati Children's Hospital Medical Center); U01HG006379 (Mayo Clinic); U01HG008679 (Geisinger Clinic); U01HG008684 (Children's Hospital of Philadelphia); U01HG008673 (Northwestern University); U01HG008701 (Vanderbilt University Medical Center serving as the Coordinating Center); U01HG008676 (Partners Healthcare/Broad Institute); U01HG008664 (Baylor College of Medicine); and U54MD007593 (Meharry Medical College). The NIDDK, NHLBI, and NHGRI had no role in the design of this study, data analysis or interpretation, or writing the manuscript.

## Author contributions

K.K. and A.G. conceived the study, provided overall supervision of the project, and made the decision to publish the findings; L.L. performed GWAS meta-analyses, conditional analyses, fine-mapping studies, functional annotations, and genetic correlation analyses. She also performed genotype quality control and primary GWAS analyses of the MESA, European, and U.S.-based discovery sub-cohorts. A.K. performed genotype quality control and GWAS analyses in the eMERGE-III sub-cohorts, and PheWAS analyses in eMERGE-III and UKBB datasets. N.Shang analyzed IgA levels derived from electronic health records. E.S. and F.Z. contributed to the analyses of genetic correlations and overlapping susceptibility with IgA nephropathy. X.Z., H.Z., J.X., and N.C. coordinated recruitment and serum IgA measurements in the Chinese GWAS cohort. H.S. coordinated recruitment and serum IgA measurements in the Japanese GWAS cohort. J.F., T.R., and F.E. recruited and characterized the German GWAS cohort. B.J., R.W., J.N., and S.H. recruited and characterized the U.S. GWAS cohort. J.Z. and S.H. managed the study database and DNA samples. P.K. performed genotype calling for microarray data. O.B., R.L., C.F., Y.L., Z.M., C.R., J.S., B.K., and J.N. performed laboratory measurements of serum IgA levels. E.O. and R.B. contributed the MESA SHARe phenotype and genotype data. G.H., C.W., M.E., T.R., J.H., H.M., L.K., B.N., T.W., R.K., S.R., E.K., S.H., J.D., I.S., and D.C. contributed eMERGE-III phenotype and genotype data. L.H., A.V., and P.M. contributed summary statistics for the Swedish discovery cohort. II consulted on the statistical analysis of genetic data. K.K., A.G., N.Steers, and J.N. provided a biological interpretation of the GWAS loci. L.L. and K.K. wrote the manuscript. All authors have read and approved the final version of the manuscript.

## Competing interests

Dr. Kiryluk has served on an advisory board for Goldfinch Bio and Gilead Sciences. Dr. Gharavi has served 1006 on an advisory board for Novartis, Travere and Natera and receives research grant funding from the Renal 1007 Research Institute and Natera. Dr. Moncrieffe is presently employed by Janssen Pharmaceutical Companies 37 1008 of Johnson & Johnson. Dr. Eitner is currently employed by Bayer AG. Drs. Julian and Novak are co-founders, 1009 co-owners of, and consultants for Reliant Glycosciences, LLC and are co-inventors on US patent application 1010 14/318,082 (assigned to UAB Research Foundation). The other authors declare no competing interests.

## Additional information

Lili Liu [1], Atlas Khan [1], Elena Sanchez-Rodriguez [1], Francesca Zanoni[1], Yifu Li[1], Nicholas Steers[1], Olivia Balderes[1], Junying Zhang [1], Priya Krithivasan[1], Robert A. LeDesma[2], Clara Fischman[3], Scott J. Hebbring[4], John B. Harley[5,6,7], Halima Moncrieffe [5,6], Leah C. Kottyan [5,6], Bahram Namjou-Khales [5,6], Theresa L. Walunas [8], Rachel Knevel[9], Soumya Raychaudhuri[9], Elizabeth W. Karlson[9], Joshua C. Denny[10], Ian B. Stanaway[11], David Crosslin[12], Thomas Rauen[13], Jürgen Floege[13], Frank Eitner[13,14], Zina Moldoveanu[15], Colin Reily[15], Barbora Knoppova [15], Stacy Hall[15], Justin T. Sheff[15], Bruce A. Julian[15], Robert J. Wyatt[16], Hitoshi Suzuki [17], Jingyuan Xie[18], Nan Chen[18], Xujie Zhou [19], Hong Zhang[19], Lennart Hammarström[20], Alexander Viktorin [21], Patrik K. E. Magnusson [21], Ning Shang [22], George Hripcsak[22], Chunhua Weng [22], Tatjana Rundek [23,24], Mitchell S. V. Elkind [25], Elizabeth C. Oelsner [1], R. Graham Barr[26,27], Iuliana Ionita-Laza[28], Jan Novak [15], Ali G. Gharavi [1] & Krzysztof Kiryluk [1] ✉

[1]Division of Nephrology, Department of Medicine, Vagelos College of Physicians & Surgeons, Columbia University, New York, NY, USA. [2]Lewis Thomas Laboratory, Department of Molecular Biology, Princeton University, Princeton, NJ, USA. [3]Department of Medicine, University of Pennsylvania, Philadelphia, PA, USA. [4]Center for Human Genetics, Marshfield Clinic Research Institute, Marshfield, WI, USA. [5]Center of Autoimmune Genomics and Etiology, Cincinnati Children's Hospital, Cincinnati, OH, USA. [6]Department of Pediatrics, University of Cincinnati College of Medicine, Cincinnati, OH, USA. [7]US Department of Veterans Affairs Medical Center, Cincinnati, OH, USA. [8]Department of Medicine, Northwestern University Feinberg School of Medicine, Chicago, IL, USA. [9]Division of Rheumatology, Immunology and Allergy, Brigham and Women's Hospital and Harvard Medical School, Boston, MA, USA. [10]Department of Medicine, Vanderbilt University School of Medicine, Nashville, TN, USA. [11]Kidney Research Institute, Division of Nephrology, Department of Medicine, University of Washington, Seattle, WA, USA. [12]Department of Biomedical Informatics and Medical Education, School of Medicine, University of Washington, Seattle, WA, USA. [13]Department of Nephrology, RWTH University of Aachen, Aachen, Germany. [14]Kidney Diseases Research, Bayer Pharma AG, Wuppertal, Germany. [15]Department of Microbiology and Medicine, University of Alabama at Birmingham, Birmingham, AL, USA. [16]Division of Pediatric Nephrology, University of Tennessee Health Sciences Center, Memphis, TN, USA. [17]Department of Nephrology, Juntendo University Faculty of Medicine, Tokyo, Japan. [18]Department of Nephrology, Institute of Nephrology, Shanghai Ruijin Hospital, Shanghai Jiao Tong University School of Medicine, Shanghai, China. [19]Renal Division, Peking University First Hospital, Peking University Institute of Nephrology, Beijing, China. [20]Department of Biosciences and Nutrition, Karolinska Institutet, Stockholm, Sweden. [21]Department of Medical Epidemiology and Biostatistics, Karolinska Institutet, Stockholm, Sweden. [22]Department of Biomedical Informatics, Vagelos College of Physicians & Surgeons, Columbia University, New York, NY, USA. [23]Department of Neurology, University of Miami, Miami, FL, USA. [24]Evelyn F. McKnight Brain Institute, University of Miami, Miami, FL, USA. [25]Department of Neurology, Vagelos College of Physicians & Surgeons, Columbia University, New York, NY, USA. [26]Division of General Medicine, Department of Medicine, Vagelos College of Physicians & Surgeons, Columbia University, New York, NY, USA. [27]Department of Epidemiology, Mailman School of Public Health, Columbia University, New York, NY, USA. [28]Department of Biostatistics, Mailman School of Public Health, Columbia University, New York, NY, USA. ✉e-mail: kk473@columbia.edu

