## [Peer Review File · Nature Communications]

Genetic regulation of serum IgA levels and susceptibility to common immune, infectious, kidney, and cardio-metabolic traitsREVIEWER COMMENTS

Reviewer #1 (Remarks to the Author):

In their paper "Genetic regulation of serum IgA levels and susceptibility to common immune, infectious, kidney and cardio-metabolic traits" the authors describe a transethnic GWAS of IgA levels that identified 9 known and 11 new loci. They carry out subsequent analyses to prioritize genes at 14 or 20 loci. They show that prioritized genes are enriched for immune defects and IgA abnormalities when manipulated in mice. They carried out genetic correlation analyses and found that IgA levels positively correlated with IgA nephropathy, type 2 diabetes, and BMI and negatively correlated with celiac disease, IBD, infections, and microbial diversity. These results will help our understanding of IGA levels and their relationship to human disease.

Some of the more novel and interesting things in the paper are that they found that IgA levels are higher in individuals of African ancestry the full significance of which is not clear but which may suggest that these higher levels conferred a fitness advantage in Africa or a fitness disadvantage in other ancestries. They also prioritized genes using colocalization methods in tissues and cell types. They examined the pathways enriched by their loci and which genes were enriched for associations in the mouse MGI phenotype database.

Questions and clarifications that could improve the paper and presentation include:

Can the authors please provide a breakdown of their hits by EAF and effect per ancestry. Are all the hits mostly coming from Europeans? Any non European hits?

In Figure 5 it looks like there is LD breakdown for IGA nephropathy vs Serum IGA levels. There may be two signals in tonsillectomy. Can the authors comment whether this is one signal that is resolving when LD breaks down across ancestries or two separate signals segregating? From conditional analyses it looks like there may be two signals only one of which associates with IgA nephropathy?

When they do the meta-PheWAS they use 1000G only to get the GPS but the study was multiethnic. Please justify.

On Sup Table 14 IgA deficiency has a negative beta when the HLA is included and a positive data when the HLA is not included. Both are statistically significant. I don't think this is an error as I see it clearly stated in another part of the paper. However, IgA deficiency should always in my mind be negatively correlated with IgA levels so I have a hard time reconciling that concept with these results. Some quirkiness of dividing the genome? The data? The statistic?

Multiple phenotypes in Sup Table 15 are seen when the HLA is included but many of these are not tested for co heritability in Sup Table 14? Examples include hypothyroidism, disorders of iron metabolism. Please expand Figure 6 A to include these or explain why these are not there. I am surprised that IgA nephropathy and IgA deficiency is not seen on the Meta-PheWAS given its strong co heritability with IgA levels. Please explain.

Sup Table 14 has a note saying Kiryluk accomplishing paper... I assume they mean accompanying paper?

In Figure 7a it looks like all SNPs have an increased association with diversity... is the whole statistic inflated. Can you please show that a random set of 20 SNPs does not have an inflated association with diversity.

In sup Figure 1 is recombination based on European ancestry? Other? Please specify.

Does the PPI network mean anything? Not discussed much.

For stepwise conditional analyses the authors say they used COJO using the same ancestries as in their discovery cohorts but these included admixed individuals. Please explain how they then derived the right mixture of samples to carry out the COJO analysis?

Please provide legends with Sup Tables. In Sup Table 3 for example I think B &H means Benjamini and Hochberg and B &Y Benjamini and Yekutieli but it would be better if that was explicitly noted.

Reviewer #2 (Remarks to the Author):

Liu et al present an association study on IgA levels, reporting 9 known and 11 novel loci. About half of the samples come from a previous Icelandic-Swedish study (Jonsson et al; Nature Genetics 2017), the rest from 16 smaller cohorts drawn from various geographic populations.

While a laudable attempt at a large-scale GWAS on IgA levels, the study has several flaws, which prevent me from recommending it for publication in its present form.

Major:

1. The authors claim that they have done a GWAS on 41,263 individuals. This is not true as 19,034 (46%) of these samples come from a previously published Icelandic-Swedish study (Jonsson et al, Nature Genetics 2017). While use of previously published data is fine, Liu et al only used the data from Supplementary Table 19 in Jonsson et al (variants with $P < 10e-6$). This table only contains 4,699 variants. Accordingly, Liu et al have not done a genome-wide analysis of 41,263 individuals in 17 cohort as they claim, but only an analysis of 4,699 variants. The authors do not disclose the number of variants that were present in all 17 cohorts (which must be 4,699 or less). As I perceive it, this reflects an attempt at deliberate obfuscation to make the study seem bigger than it is, which is of course unacceptable. Please be accurate, truthful, and do not exaggerate. The authors need to disclose the number of variants genotyped in each cohort, and how many of these were present in all 17 cohorts (or 16 cohorts if the Icelandic-Swedish cohort is omitted). It may be that they did a true genome-wide scan across 22,229 individuals from 16 cohorts, but then that should be clearly stated.
2. Similarly, since the Icelandic-Swedish study constitute such a big part of the study, the fact that Liu et al find 9 known loci is expected, as the majority of known IgA variants come from Jonsson et al. This circularity needs to be commented.
3. The authors used fixed-effects meta-analysis to combine association statistics from different ethnicities. This is not correct, as the effects of sequence variants on IgA levels may vary between ethnicities. Fixed-effects may over-estimate the effects of a variant in this situation. The authors should instead do random-effects meta-analysis to account for such variation in effects (this is also done in most recent trans-ethnic association studies).
4. Heterogeneity tests for the fixed-effects meta-analysis are not reported. The authors need to add Cochran's Q and I² statistics, and comment any significant heterogeneity.
5. Similarly, for novel variants, the association statistics in each cohort should be reported. The reader should be able to assess if the effects are consistent across cohorts, or mainly driven by a few cohorts. This is particularly important for new low-frequency variants, such as the GPATCH2 variant.
6. The authors perform conditional analysis using summary statistics and an external reference panel (for LD calculation). This dubious practice, and raises question marks around the "conditionally

independent signals”, particularly as these were not detected in the Swedish-Icelandic study which again constitutes 46% of the data in the current study. The authors should do conditional analysis in the original cohorts, using the individual-level data.

7. The authors text describe known loci at length (e.g., ELL2, ANKRD55, etc). These sections should be removed. They have been analyzed and described previously, and the authors should not attempt to give the impression that they are novel. It would be more appropriate to limit the text to the novel loci.

Minor:

8. Introduction: “...diverse ancestries, maximizing the power for genetic discovery”. This is not true. A genetically heterogenous study population, or a multi-ethnic meta-analysis, does not maximize the power to detect genetic associations. It will help identifying variants with consistent effects across many ethnicities, but will make it more difficult to identify rare variants with strong effects (which are often population-specific).

9. Latinx is a neologism, the use of which is not common practice.

REVIEWER COMMENTS

We would like to thank the reviewers for their thoughtful comments and suggestions. In response, we have restructured the manuscript and refined some of our analyses. Overall, we believe the manuscript is considerably strengthened by the revisions suggested by the reviewers. Please see our detailed responses below:

Reviewer #1 (Remarks to the Author):

In their paper “Genetic regulation of serum IgA levels and susceptibility to common immune, infectious, kidney and cardio-metabolic traits” the authors describe a transethnic GWAS of IgA levels that identified 9 known and 11 new loci. They carry out subsequent analyses to prioritize genes at 14 or 20 loci. They show that prioritized genes are enriched for immune defects and IgA abnormalities when manipulated in mice. They carried out genetic correlation analyses and found that IgA levels positively correlated with IgA nephropathy, type 2 diabetes, and BMI and negatively correlated with celiac disease, IBD, infections, and microbial diversity. These results will help our understanding of IGA levels and their relationship to human disease.

Some of the more novel and interesting things in the paper are that they found that IgA levels are higher in individuals of African ancestry the full significance of which is not clear but which may suggest that these higher levels conferred a fitness advantage in Africa or a fitness disadvantage in other ancestries. They also prioritized genes using colocalization methods in tissues and cell types. They examined the pathways enriched by their loci and which genes were enriched for associations in the mouse MGI phenotype database.

Questions and clarifications that could improve the paper and presentation include:

Can the authors please provide a breakdown of their hits by EAF and effect per ancestry. Are all the hits mostly coming from Europeans? Any non-European hits?

We agree, and we now generated forest plots for meta-analyses defined by ancestry for all 20 significant loci. We included EAF, OR and 95% CIs information separately for each major ancestral group included in our study. We displayed the previously published DeCode cohort (100% North European) separately, for reference, as also suggested by the second reviewer (see new Supplementary Figures 4 and 5). For example, these plots clearly illustrate that the *RUNX3* locus is supported only by European ancestry cohorts, while *GPATCH2* locus is supported only by African and Admixed ancestry cohorts; we hope this presentation provides the requested information. In addition, we generated a table showing summary statistics for each individual cohort, including ORs, SEs, and P-values (see new Supplementary Table 1). To strengthen our initial observations of ancestral differences in IgA levels, we have now added comparisons of polygenic risk score for IgA levels between major reference populations of 1000G. Strikingly, the mean GPS in the AFR population was over 2 standard deviations higher compared to the EUR population. Moreover, for the top 1% variants we observed a significant positive rank correlation between AFR-EUR allelic frequency difference and effect size. Please refer to the updated Figure 1 for details of these new analyses.

In Figure 5 it looks like there is LD breakdown for IGA nephropathy vs Serum IGA levels. There may be two signals in tonsillectomy. Can the authors comment whether this is one signal that is resolving when LD breaks down across ancestries or two separate signals segregating? From conditional analyses it looks like there may be two signals only one of which associates with IgA nephropathy?

Thank you for this comment. As also suggested by the second reviewer, we have now re-performed stepwise conditional analyses for all our GWAS loci using primary genotype data (all new cohorts) combined with the COJO-based conditioning of the DECODE and Swedish cohorts (1000G Europeans used for LD reference). These results indicated only one independent signal at the *HORMAD2/LIF* locus. Similarly, only one independent association signal at this locus was identified for IgA nephropathy. The pattern suggestive of the apparent LD breakdown results from partial SNP coverage of this locus in the DECODE cohort and, for this reason, in our colocalization studies we removed the DECODE study. The IgA level signal (without DECODE) colocalized with the signal for IgA nephropathy (PP4=0.98) and tonsillectomy (PP4=0.67). To reduce confusion, and for consistency with our co-localization analyses, we have

now revised the regional plot for IgA levels in Figure 5 by removing the DECODE cohort from the meta-analysis (see new Figure 5).

When they do the meta-PheWAS they use 1000G only to get the GPS but the study was multiethnic. Please justify.

We used 1000G reference including all populations except for the South Asian population, since South Asians were not represented in our study. This reference panel is publicly available and has been used commonly in GPS modeling, and we made sure to include all populations representative of our GWAS participants. Unfortunately, more diverse reference panels such as TopMed are not publicly available for download, precluding their use for this purpose. We have now clarified this in the methods section under the Meta-PheWAS section. We also added the discussion of the limitations of this approach.

On Sup Table 14 IgA deficiency has a negative beta when the HLA is included and a positive data when the HLA is not included. Both are statistically significant. I don't think this is an error as I see it clearly stated in another part of the paper. However, IgA deficiency should always in my mind be negatively correlated with IgA levels so I have a hard time reconciling that concept with these results. Some quirkiness of dividing the genome? The data? The statistic?

Thank you for this comment. You are correct, this is not an error, and we think this finding is reflective of true biology. IgA deficiency has an established auto-immune component, but the exact pathogenesis is poorly understood. The negative correlation induced by the HLA region is likely reflective of distinctly different classical HLA alleles driving IgA production vs. autoimmune processes leading to IgA deficiency. At the same time, the observed positive genetic correlation at non-HLA loci suggests that some of these loci are shared between the two conditions. Interestingly, positive correlation of IgA deficiency has been previously reported with allergic diseases, and we also observed positive correlation between IgA levels and allergy, suggesting that all three traits may share genetic causes outside of the HLA region. Obviously, more work will be needed to dissect these shared genetic mechanisms.

Multiple phenotypes in Sup Table 15 are seen when the HLA is included but many of these are not tested for co heritability in Sup Table 14? Examples include hypothyroidism, disorders of iron metabolism. Please expand Figure 6A to include these or explain why these are not there. I am surprised that IgA nephropathy and IgA deficiency is not seen on the Meta-PheWAS given its strong co heritability with IgA levels. Please explain.

For genetic correlation analyses, we were able to only include the conditions and traits that 1) were previously studied by GWAS, 2) had significant SNP-based heritability, and 3) had publicly available GWAS summary statistics available for analysis. This explains why we were unable to evaluate all of the conditions picked up by Meta-PheWAS. Conversely, the lack of Meta-PheWAS association with some conditions is explained by the limitations of population-based biobanks and ICD coding schemes. For example, IgA nephropathy is not well captured by specific ICD codes, and as a result no specific phecode for IgA nephropathy presently exists (in fact, this represents a major problem for biobank-based studies of IgA nephropathy). Reassuringly, we do observe a positive association with hematuria, the primary symptom of IgA nephropathy. For IgA deficiency, this condition is less prevalent, under-diagnosed, and maps to a less specific phecode "Deficiency of humoral immunity", potentially explaining why this association would be missed in biobank-based PheWAS. We now mention these limitations in the discussion section.

Sup Table 14 has a note saying Kiryluk accomplishing paper... I assume they mean accompanying paper?

Sorry for the typo, we have now corrected this in the revised manuscript.

In Figure 7a it looks like all SNPs have an increased association with diversity... is the whole statistic inflated. Can you please show that a random set of 20 SNPs does not have an inflated association with diversity.

We are particularly grateful for this astute comment since it helped us to identify a problem with our input datasets. Your comment prompted us to re-examine the summary statistics of the published microbiome GWAS used in our analysis, and indeed we uncovered substantial genomic inflation specifically in the GWAS for microbiome diversity ($\lambda=4.82!$). Although the statistics for individual species did not exhibit similar inflation, we now decided to

completely remove the microbiome analysis from the manuscript, since this finding raised questions about the validity of the results. Again, we greatly appreciate your help in identifying this issue, which we have clearly overlooked in our initial analysis scripts that parsed through a large number of GWAS summary statistics for different microbiome features.

In sup Figure 1 is recombination based on European ancestry? Other? Please specify.

Because our GWAS is multiethnic, we used the recombination rates generated based on all 1000G populations. We added this information to the legend of Supplementary Figure 1 and 3.

Does the PPI network mean anything? Not discussed much.

Thanks, we have now added the description of the PPI network analysis to the results section. The key finding is that our positional candidate genes across all loci were more likely to have direct protein-protein interactions than expected by chance, suggesting that they participate in common biological processes. We have now added this information before discussing pathway enrichments.

For stepwise conditional analyses the authors say they used COJO using the same ancestries as in their discovery cohorts but these included admixed individuals. Please explain how they then derived the right mixture of samples to carry out the COJO analysis?

Thank you for pointing out the limitations of using COJO in GWAS of admixed populations. We completely agree, and we have now re-performed stepwise conditional analysis for each genome-wide significant locus by conditioning using primary genotype data instead of COJO. We use COJO only for the two previously published European cohorts (Decode and Swedish) for which we do not have primary genotype data; because these cohorts are 100% European and do not include admixed individuals, we used 1000G Europeans for LD reference. We believe that this strategy provides us with more accurate results. Table 2 and the manuscript results have now been revised accordingly.

Please provide legends with Sup Tables. In Sup Table 3 for example I think B &H means Benjamini and Hochberg and B &Y Benjamini and Yekutieli but it would be better if that was explicitly noted.

Thank you for pointing this out, we have now expanded the legends in the Supplementary Tables as requested.

Reviewer #2 (Remarks to the Author):

Liu et al present an association study on IgA levels, reporting 9 known and 11 novel loci. About half of the samples come from a previous Icelandic-Swedish study (Jonsson et al; Nature Genetics 2017), the rest from 16 smaller cohorts drawn from various geographic populations.

While a laudable attempt at a large-scale GWAS on IgA levels, the study has several flaws, which prevent me from recommending it for publication in its present form.

Major:

1. The authors claim that they have done a GWAS on 41,263 individuals. This is not true as 19,034 (46%) of these samples come from a previously published Icelandic-Swedish study (Jonsson et al, Nature Genetics 2017). While use of previously published data is fine, Liu et al only used the data from Supplementary Table 19 in Jonsson et al (variants with $P < 10e-6$). This table only contains 4,699 variants. Accordingly, Liu et al have not done a genome-wide analysis of 41,263 individuals in 17 cohort as they claim, but only an analysis of 4,699 variants. The authors do not disclose the number of variants that were present in all 17 cohorts (which must be 4,699 or less). As I perceive it, this reflects an attempt at deliberate obfuscation to make the study seem bigger than it is, which is of course unacceptable. Please be accurate, truthful, and do not exaggerate. The authors need to disclose the number of variants genotyped in each cohort, and how many of these were present in all 17 cohorts (or 16 cohorts if the Icelandic-Swedish cohort is omitted). It may be

that they did a true genome-wide scan across 22,229 individuals from 16 cohorts, but then that should be clearly stated.

We apologize if our initial presentation was not clear, it was certainly not our intention to obfuscate. We have now revised the manuscript to more clearly present the design and composition of our study. The authors of the DeCode paper declined to release their published summary statistics, limiting our analysis to only suggestive SNPs for that cohort. All other cohorts were genotyped genome-wide, as described in detail under individual cohort descriptions (see Methods). These descriptions now also include the exact number of variants included in GWAS after all post-imputation QC filters. The issue of uneven genomic coverage is now also discussed in detail in the revised limitations section. We hope these changes adequately address your concerns.

2. Similarly, since the Icelandic-Swedish study constitute such a big part of the study, the fact that Liu et al find 9 known loci is expected, as the majority of known IgA variants come from Jonsson et al. This circularity needs to be commented.

Again, we apologize for our confusing presentation. The key point is that our study provides an independent replication of most known loci in the new ancestrally diverse cohorts. In the combined meta-analysis, these known signals became more significant compared to the original reports, providing reassurance that these loci represent true positives. We have now revised the text accordingly, and we clarified our validation of these loci. Moreover, some of the suggestive loci from the DECODE analysis now become genome-wide significant in the combined meta-analyses, providing strong evidence for their involvement in the determination of serum IgA levels. The newly added forest plots (see Supplementary Figures 4 and 5) provide a clear breakdown of signal contributions from DECODE and non-DECODE cohorts for each of the 20 genome-wide significant loci.

3. The authors used fixed-effects meta-analysis to combine association statistics from different ethnicities. This is not correct, as the effects of sequence variants on IgA levels may vary between ethnicities. Fixed-effects may over-estimate the effects of a variant in this situation. The authors should instead do random-effects meta-analysis to account for such variation in effects (this is also done in most recent trans-ethnic association studies).

Thank you for this comment, we agree that this is especially important for the loci exhibiting significant heterogeneity by ancestry. At the suggestion of the reviewer, we have now added random effects meta-analysis using TransMeta, a method that is specifically designed to handle multi-ancestry analysis¹. The random effects statistics were added in the revised Table 2. Notably, the results of TransMeta showed that all 20 genome-wide significant loci identified using fixed-effect model were also genome-wide significant under the TransMeta random effects model. We also detected no significant heterogeneity by Cochran's heterogeneity test, further suggesting that fixed effects meta-analysis is appropriate. The new results for both models and the heterogeneity tests are now included in the updated Table 2.

4. Heterogeneity tests for the fixed-effects meta-analysis are not reported. The authors need to add Cochran's Q and I² statistics and comment any significant heterogeneity.

The heterogeneity statistics have now been added to the revised Table 2 as suggested.

5. Similarly, for novel variants, the association statistics in each cohort should be reported. The reader should be able to assess if the effects are consistent across cohorts, or mainly driven by a few cohorts. This is particularly important for new low-frequency variants, such as the GPATCH2 variant.

Thank you for this suggestion. we have now generated a table showing the statistics for each cohort, including OR, SE, and P-values, and the consistent effects across cohorts were highlighted in red color. In addition, we generated forest plots by ancestry, including OR, 95% CIs and effect allele frequency for each major ancestral group. For comparison, we kept the Decode cohort as a separate entry in these plots. The forest plots demonstrate that there are some signals clearly driven by non-European ancestry cohorts, such as the GPATCH2 locus predominately driven by African and Admixed ancestry cohorts. The forest plots and the cohort-specific results have now been added as Supplementary Figures 4 & 5, and a Supplementary Table 1, respectively.

6. The authors perform conditional analysis using summary statistics and an external reference panel (for LD

calculation). This dubious practice, and raises question marks around the “conditionally independent signals”, particularly as these were not detected in the Swedish-Icelandic study which again constitutes 46% of the data in the current study. The authors should do conditional analysis in the original cohorts, using the individual-level data.

Thank you for this helpful suggestion. As described in our response to the first reviewer, we have now re-performed stepwise conditional analyses for each genome-wide significant locus using primary genotype data. We apply COJO-based analysis using European 1000G reference only to the Decode and Swedish cohorts for which we do not have primary genotype data. We have revised Table 2 and the entire manuscript accordingly.

7. The authors text describe known loci at length (e.g., ELL2, ANKRD55, etc). These sections should be removed. They have been analyzed and described previously, and the authors should not attempt to give the impression that they are novel. It would be more appropriate to limit the text to the novel loci.

Our goal was to perform uniformly comprehensive genomic annotations for all known and novel loci to systematically prioritize potential causal genes. Similar functional annotations using the latest functional genomic resources have not been performed in the same fashion for some of the older loci. Indeed, some of the newly prioritized genes at the known loci clearly fall within the same signaling pathways as the genes prioritized at novel loci based on our analyses. For this reason, we prefer to retain new annotations of both new and old loci, since some of our annotations enhance the interpretation of new loci (e.g., *RUNX2* and *RUNX3*). However, we have now re-organized the text, placing considerably more emphasis on the novel loci, and shortening the text for the known loci. Throughout the text (and in the tables), we make every attempt at indicating which loci are novel and which loci have been previously reported.

Minor:

8. Introduction: “...diverse ancestries, maximizing the power for genetic discovery”. This is not true. A genetically heterogenous study population, or a multi-ethnic meta-analysis, does not maximize the power to detect genetic associations. It will help identifying variants with consistent effects across many ethnicities, but will make it more difficult to identify rare variants with strong effects (which are often population-specific).

Thank you for pointing this out since this was not our intended meaning of this sentence (we meant to say that we were maximizing the sample size for genetic discovery by aggregating all available cohorts). We have now deleted this statement altogether.

9. Latinx is a neologism, the use of which is not common practice.

Thank you for this comment. The gender-neutral term “Latinx” has become the preferred term over “Latino/Latina” or “Hispanic” when referring to the ethnicity of the admixed Caribbean population. This term has entered the medical literature only recently but is an official entry in the Encyclopedia Britannica and the Merriam-Webster dictionary, also now widely used in over 2,400 PubMed-indexed papers in the past 3 years. Therefore, we prefer to keep this term if possible.

References:

1. Shi, J. & Lee, S. A novel random effect model for GWAS meta-analysis and its application to trans-ethnic meta-analysis. *Biometrics* **72**, 945-54 (2016).

REVIEWERS' COMMENTS

Reviewer #1 (Remarks to the Author):

My concerns have been addressed.

Reviewer #2 (Remarks to the Author):

The authors have done a good job reviewing the manuscript.

I only find one remaining detail that needs to be corrected before the paper can be accepted:

Line 90, "by DeCode Genetics" is not completely accurate. Please correct to "by deCODE Genetics and Lund University". The two institutions jointly directed the work, generated independent datasets, and share the last-authorship (see author list and Methods section of PMID 28628107).

Line 118: Same.

Line 426: Same.

Line 429: Same.

Line 431: Same.

Line 536: Same.

Line 548: Same.

Line 568: Correct "DeCode and Swedish" to "Icelandic and Swedish".

Line 1007: Correct "Decode" to "Icelandic and Swedish".

REVIEWERS' COMMENTS

Reviewer #1 (Remarks to the Author):

My concerns have been addressed.

Reviewer #2 (Remarks to the Author):

The authors have done a good job reviewing the manuscript. I only find one remaining detail that needs to be corrected before the paper can be accepted: Line 90, "by DeCode Genetics" is not completely accurate. Please correct to "by deCODE Genetics and Lund University". The two institutions jointly directed the work, generated independent datasets, and share the last authorship (see author list and Methods section of PMID 28628107).

Line 118: Same.

Line 426: Same.

Line 429: Same.

Line 431: Same.

Line 536: Same.

Line 548: Same.

Line 568: Correct "DeCode and Swedish" to "Icelandic and Swedish".

Line 1007: Correct "Decode" to "Icelandic and Swedish".

Response:

Thank you for these comments. We have now made these corrections as suggested by the second reviewer.